# XPC–PARP complexes engage the chromatin remodeler ALC1 to catalyze global genome DNA damage repair

Charlotte Blessing [1,2,13], Katja Apelt[3,13], Diana van den Heuvel[3], Claudia Gonzalez-Leal[1,2], Magdalena B. Rother[3], Melanie van der Woude[4], Román González-Prieto [5,6,7], Adi Yifrach[8], Avital Parnas[8], Rashmi G. Shah[9], Tia Tyrsett Kuo [1,2], Daphne E. C. Boer[3], Jin Cai[1,2], Angela Kragten[3], Hyun-Suk Kim[10], Orlando D. Schärer[10,11], Alfred C. O. Vertegaal[5], Girish M. Shah [9], Sheera Adar[8], Hannes Lans [4], Haico van Attikum[3], Andreas G. Ladurner [1,2,12,14✉] & Martijn S. Luijsterburg [3,14✉]

Cells employ global genome nucleotide excision repair (GGR) to eliminate a broad spectrum of DNA lesions, including those induced by UV light. The lesion-recognition factor XPC initiates repair of helix-destabilizing DNA lesions, but binds poorly to lesions such as CPDs that do not destabilize DNA. How difficult-to-repair lesions are detected in chromatin is unknown. Here, we identify the poly-(ADP-ribose) polymerases PARP1 and PARP2 as constitutive interactors of XPC. Their interaction results in the XPC-stimulated synthesis of poly-(ADP-ribose) (PAR) by PARP1 at UV lesions, which in turn enables the recruitment and activation of the PAR-regulated chromatin remodeler ALC1. PARP2, on the other hand, modulates the retention of ALC1 at DNA damage sites. Notably, ALC1 mediates chromatin expansion at UV-induced DNA lesions, leading to the timely clearing of CPD lesions. Thus, we reveal how chromatin containing difficult-to-repair DNA lesions is primed for repair, providing insight into mechanisms of chromatin plasticity during GGR.

[1] Biomedical Center (BMC), Physiological Chemistry, Faculty of Medicine, LMU Munich, Planegg-Martinsried, Germany. [2] International Max Planck Research School (IMPRS) for Molecular Life Sciences, Planegg-Martinsried, Germany. [3] Department of Human Genetics, Leiden University Medical Center (LUMC), Leiden, The Netherlands. [4] Department of Molecular Genetics, Erasmus MC Cancer Institute, Erasmus University Medical Center, Rotterdam, The Netherlands. [5] Department of Cell and Chemical Biology, Leiden University Medical Center (LUMC), Leiden, The Netherlands. [6] Genome Proteomics Laboratory, Andalusian Center For Molecular Biology and Regenerative Medicine (CABIMER), University of Seville, Seville, Spain. [7] Department of Cell Biology, University of Seville, Seville, Spain. [8] Department of Microbiology and Molecular Genetics, The Institute for Medical Research Israel-Canada, The Faculty of Medicine, The Hebrew University of Jerusalem, Jerusalem, Israel. [9] Laboratory for Skin Cancer Research, CHU-Q: Laval University Hospital Research Centre of Quebec (CHUL site), Quebec City, Canada. [10] Center for Genomic Integrity, Institute for Basic Science, Ulsan, Republic of Korea. [11] Department of Biological Sciences, School of Life Sciences, Ulsan National Institute of Science and Technology, Ulsan, Republic of Korea. [12] Eisbach Bio GmbH, Planegg-Martinsried, Germany. [13] These authors contributed equally: Charlotte Blessing, Katja Apelt. [14] These authors jointly supervised this work: Andreas G. Ladurner, Martijn S. Luijsterburg. ✉email: andreas.ladurner@bmc.med.lmu.de; m.luijsterburg@lumc.nl

The integrity of the human genome is constantly threatened by endogenous and exogenous sources, which cause up to $10^5$ DNA lesions per cell per day[1]. Cells thus critically depend on the accuracy of dedicated DNA repair mechanisms to recognize and remove genomic DNA lesions and maintain genome integrity[2].

One principal source of DNA damage is UV light, which results in the crosslinking of neighboring bases on the same DNA strand, forming so-called 6-4 photoproducts (6-4PPs) and cyclobutane pyrimidine dimers (CPDs). Nucleotide excision repair (NER) can repair these bulky lesions[3]. The initiation of this repair pathway depends on the position of the lesion in the genome. While RNA polymerase II stalling at lesions in transcribed strands initiates transcription-coupled repair (TCR or TC-NER)[4,5], lesions in transcriptionally inactive genome regions are recognized by specialized damage sensors that initiate global genome repair (GGR or GG-NER)[6,7]. Recognition through both sub-pathways ultimately leads to a common pathway of verification, excision, and re-synthesis of the damaged DNA. In principle, all core NER factors have been identified and the fundamental DNA repair process can be reconstituted in vitro[8,9]. However, the cellular repair mechanism is not well understood in the chromatin context, as knowledge about chromatin factors that allow and promote the efficient action of the core repair factors is limited.

In mammalian cells, GGR is initiated by XPC, which forms a complex with RAD23B and CEN2[6,10–12]. Rather than binding to lesions directly, XPC binds the accessible, non-damaged DNA that is opposite the DNA injury. This allows the recognition of a broad spectrum of lesions that are structurally unrelated[13,14]. XPC binding results in the slight opening of the DNA surrounding a lesion (~6 nucleotides)[15], which facilitates the binding of subsequent factors. The recruitment of these downstream NER proteins, including TFIIH, RPA, XPA, XPG, and ERCC1-XPF, to sites of UV-induced DNA lesions is abolished in XPC-deficient cells. This demonstrates that the GGR repair pathway is strictly dependent on XPC[10].

Although XPC has a high affinity for 6-4PPs, it's binding to CPDs is rather inefficient due to minimal thermodynamic helix destabilization caused by the latter lesion. The recognition and repair of CPDs, therefore, requires the additional action of the damaged DNA-binding protein 2[16], which does not seem to influence in vitro reconstituted NER[8,17]. DDB2 further utilizes slide-assisted site exposure to detect inaccessible lesions occluded in nucleosomes[18], and creates a local chromatin environment around lesions that facilitates the assembly of repair complexes[19–22]. DDB2 is thus often considered the factor that prepares chromatin for GGR, and its dissociation from DNA lesions is subsequently required for the progression of repair[23]. Whether this is an exclusive feature of DDB2 or whether and how XPC also contributes to local chromatin changes is unknown.

In addition to XPC and DDB2, PARP1 is also known to associate with UV-induced DNA lesions resulting in poly-(ADP-ribos)ylation (PARylation) at sites of DNA damage[24–26]. In vitro approaches showed that PARP1 and DDB2 can simultaneously bind to a UV-induced CPD[25]. In agreement, in situ fractionation showed that endogenous PARP1 is recruited to sites of local UV damage[24,25,27]. However, the precise interplay between the three lesion-recognition proteins XPC, DDB2 and PARP1 during GGR is poorly understood. Initially, DDB2 was found to associate with and potentially stimulate the catalytic activity of PARP1, resulting in the PARylation of DDB2. This was suggested to counteract DDB2 auto-ubiquitylation and its subsequent degradation[28]. Consistently, inhibition of PARP activity was found to accelerate DDB2 degradation[28] and to reduce XPC recruitment to UV lesions under conditions of low damage load[20,29]. PARP1 was

also shown to stimulate XPC recruitment to DNA damage sites in a DDB2-independent manner and to regulate XPC release[24]. Thus, it appears that both DDB2 and PARP1 may stimulate XPC recruitment to initiate GGR. However, the relevance of the poly-(ADP-ribose) response for GGR and how it is related to XPC activity remains to be established.

In addition to PARP1, the nuclear and DNA damage-dependent PARP2 enzyme seems to have independent functions in the DNA damage response, which are not well understood. In general, PARP1 is thought to provide ~90% of the PAR signal at DNA lesions, while PARP2 contributes a minor part[30,31]. Instead, PARP2 is suggested to increase the branching of PAR chains[32]. The double knockout of both PARP1 and PARP2 is embryonic lethal in mice[33] and renders cells highly sensitive to DNA-damaging agents, such as the alkylating agent methyl methanesulfonate[34]. This suggests that both proteins cooperatively act in the DNA damage response. However, whether PARP2 contributes to the poly-(ADP-ribose) response and has roles in GGR remains to be established.

Here, we sought to elucidate the role of PARylation, PARP2, and active chromatin remodeling in GGR. Several chromatin remodelers have been implicated in DNA repair, notably ALC1, which acts downstream of PARP1/2 activation, due to its strict PARylation-dependent nucleosome remodeling activity[35–38]. Using proteomics, live-cell imaging, and UV-induced DNA damage, our data revealed a new XPC-PARP axis that links ATP-dependent and PARylation-activated ALC1-mediated chromatin remodeling to GGR. We identify the poly-(ADP-ribose) polymerases PARP1 and PARP2 as constitutive interactors of the damage-recognition protein XPC. The close interaction between these proteins results in an XPC-dependent stimulation of the poly-(ADP-ribose) response, which facilitates the recruitment of the poly-(ADP-ribose)-dependent chromatin remodeler ALC1. We thus identify a new XPC-dependent mechanism that impacts the chromatin environment and promotes chromatin remodeling at UV lesions.

## Results

**XPC interacts with PARP1 and PARP2.** To identify potential new factors involved in GGR, we analyzed the interactome of the damage-recognition factor XPC by mass spectrometry. We generated a knockout (KO) of XPC in U2OS (FRT) cells containing an FRT Flp-In integration site. Having confirmed the successful knockout of XPC by sequencing and western blot analysis (Supplementary Fig. 1a), we exploited the site-specific transgene integration of the Flp-In system to re-express XPC-GFP in these cells under a doxycycline-inducible promoter. To demonstrate the functionality of our newly generated cell system, we measured the ability to repair UV lesions in unscheduled DNA synthesis (UDS) assays. While the XPC-KO showed a severe repair defect, the re-expression XPC-GFP restored the capacity of the cells to repair UV-induced DNA damage (Supplementary Fig. 1b, c). Label-free proteomics after pull-down of XPC-GFP revealed several known XPC-binding proteins, including RAD23A/B and CEN2/3, as the top interactors (Fig. 1a). After UV irradiation, XPC additionally became tightly bound to several GGR factors, such as the DDB2 complex (containing DDB1 and CUL4A/B), the TFIIH subunits GTF2H1-4 (p62, p44, p34, p52), XPB (p89/ERCC3) and XPD (p80/ERCC2) (Fig. 1b). This confirms that our label-free proteomics approach is suitable to detect interactions within an active GGR process.

Interestingly, our analysis further identified the poly-(ADP-ribose) polymerases PARP1 and PARP2 as strong XPC-associated proteins (Fig. 1a). In contrast to the main GGR factors, the interaction between XPC and PARP1/2 was not significantly

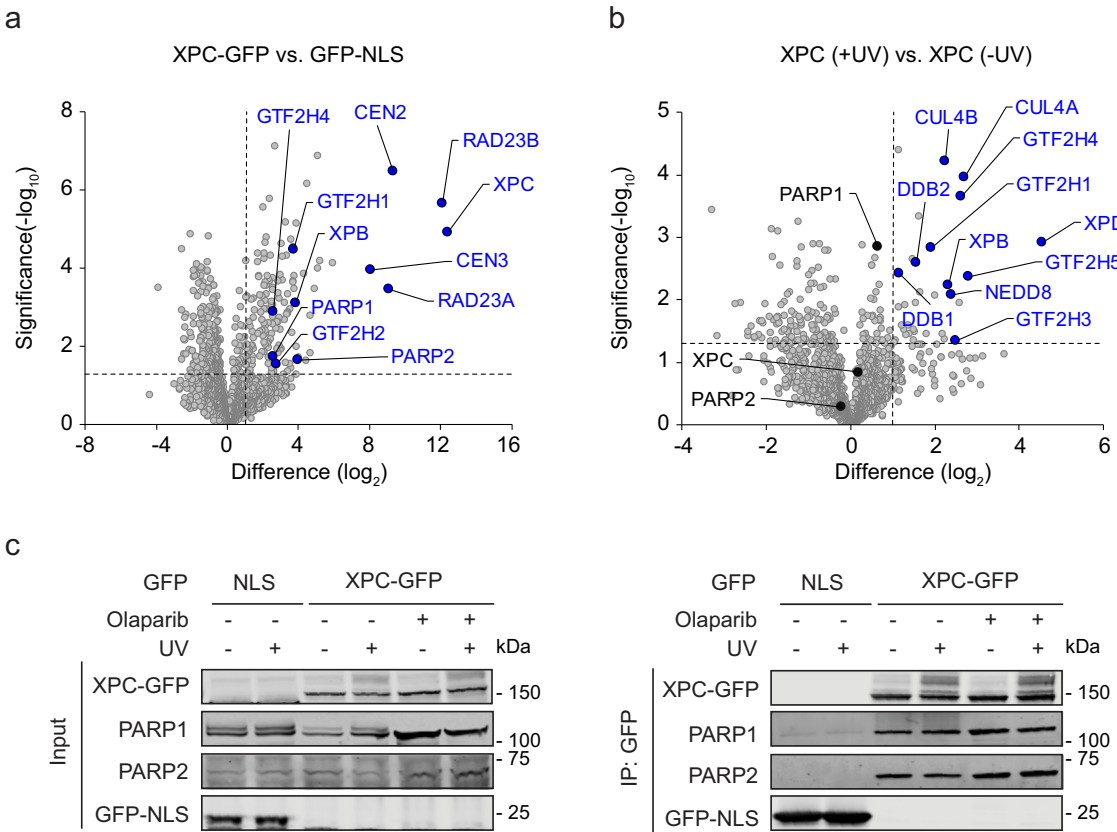

**Fig. 1 XPC interacts with PARP1 and PARP2 in a UV-independent manner. a** Volcano plot displaying the interactome of XPC-GFP over GFP-NLS after GFP-pull-down from U2OS (FRT) XPC-KO cells and analysis by label-free proteomics. **b** Differential interactome of XPC-GFP comparing UV-C-irradiated (20 J/m², 1 h) vs. unirradiated U2OS (FRT) XPC-KO cells. **a, b** The dashed lines indicate a twofold enrichment on the x axis (log₂ of 1) and a significance of 0.05 (−log₁₀ P value of 1.3; two-sided t test) on the y axis. **c** Co-Immunoprecipitation (Co-IP) of GFP-NLS and XPC-GFP in the presence and absence of UV-C (20 J/m², 1 h) and the PARP inhibitor olaparib (10 μM). Three independent replicates of each IP experiment were performed obtaining similar results.

affected by UV irradiation (Fig. 1b). Immunoprecipitation experiments confirmed that the XPC-PARP1/2 interaction was constitutive and independent of PARylation, as treatment with the PARP inhibitor olaparib did not affect the interactions (Fig. 1c). Our findings indicate that XPC forms a constitutive interaction with both PARP1 and PARP2, which is not affected by DNA damage or PARylation.

**XPC and ALC1 interact more strongly with PARP2 than PARP1.** To obtain first insights into the role of PARP1 and PARP2 in GGR and the relevance of their interaction with XPC, we conducted an orthogonal experiment to identify the interactome of the two PARP enzymes. We stably expressed PARP1-GFP or GFP-PARP2 in U2OS (FRT) cells (Supplementary Fig. 1d) and performed label-free proteomics after GFP-pull-down of the tagged proteins. PARP1 most abundantly interacted with XRCC1-LIG3 and POLB, and further showed robust interactions with PARP2, histones, as well as the known poly-(ADP-ribose)-binding proteins ALC1 and macroH2A (Fig. 2a). In contrast, XPC was not significantly enriched in the PARP1 interactome (Fig. 2a, b), which is likely caused by the high abundance of PARP1 and a potentially low stoichiometric interaction with XPC. It should be noted that earlier studies did report an interaction between immunoprecipitated endogenous PARP1 and XPC[24].

Interestingly, the interactome of PARP2 revealed the poly-(ADP-ribose)-dependent chromatin remodeler ALC1 as its most abundant interactor. PARP2 further interacted with XRCC1-LIG3, PARP1, histones, and macroH2A (Fig. 2c). PARP2 also

clearly interacted with XPC in a manner that was not affected by UV irradiation (Fig. 2c, d). Intensity-based absolute quantification (iBAQ) of protein amounts indicated that ~15% of the isolated PARP2 molecules were associated with ALC1, while only 0.07% of PARP1 molecules interacted with the remodeler. Additionally, the fraction of PARP2 molecules associated with XPC was ten-fold higher than for PARP1 (Fig. 2e). The stoichiometry among PARP enzymes was very low with 1.76% of the PARP2 molecules interacting with PARP1, ruling out indirect interactions through PARP heterodimerization. Immunoprecipitation experiments confirmed that PARP2 robustly interacted with both ALC1 and XPC, while these interactions were not or only weakly detected after pull-down of PARP1 (Fig. 2f). This demonstrates that XPC and ALC1 both preferentially associate with PARP2 over PARP1 under our experimental conditions.

**PARP1/2 recruitment to UV lesions is independent of XPC.** The DNA damage-recognition proteins XPC and DDB2 can both bind to UV-induced lesions. DDB2 thereby stimulates XPC recruitment at difficult-to-detect lesions, such as CPDs[16,20,39], and facilitates lesion-recognition by XPC in a chromatin context. PARP1 was also reported to bind to UV-induced DNA lesions together with XPC[24,25]. Considering the interaction between XPC and PARP1/2, we asked whether the PARP enzymes are recruited to UV lesions and if this is stimulated by XPC. To this end, we employed a UV-C (266 nm) laser live-cell imaging set-up, in which all optics have been replaced by quartz glass. Local irradiation with UV-C led to the

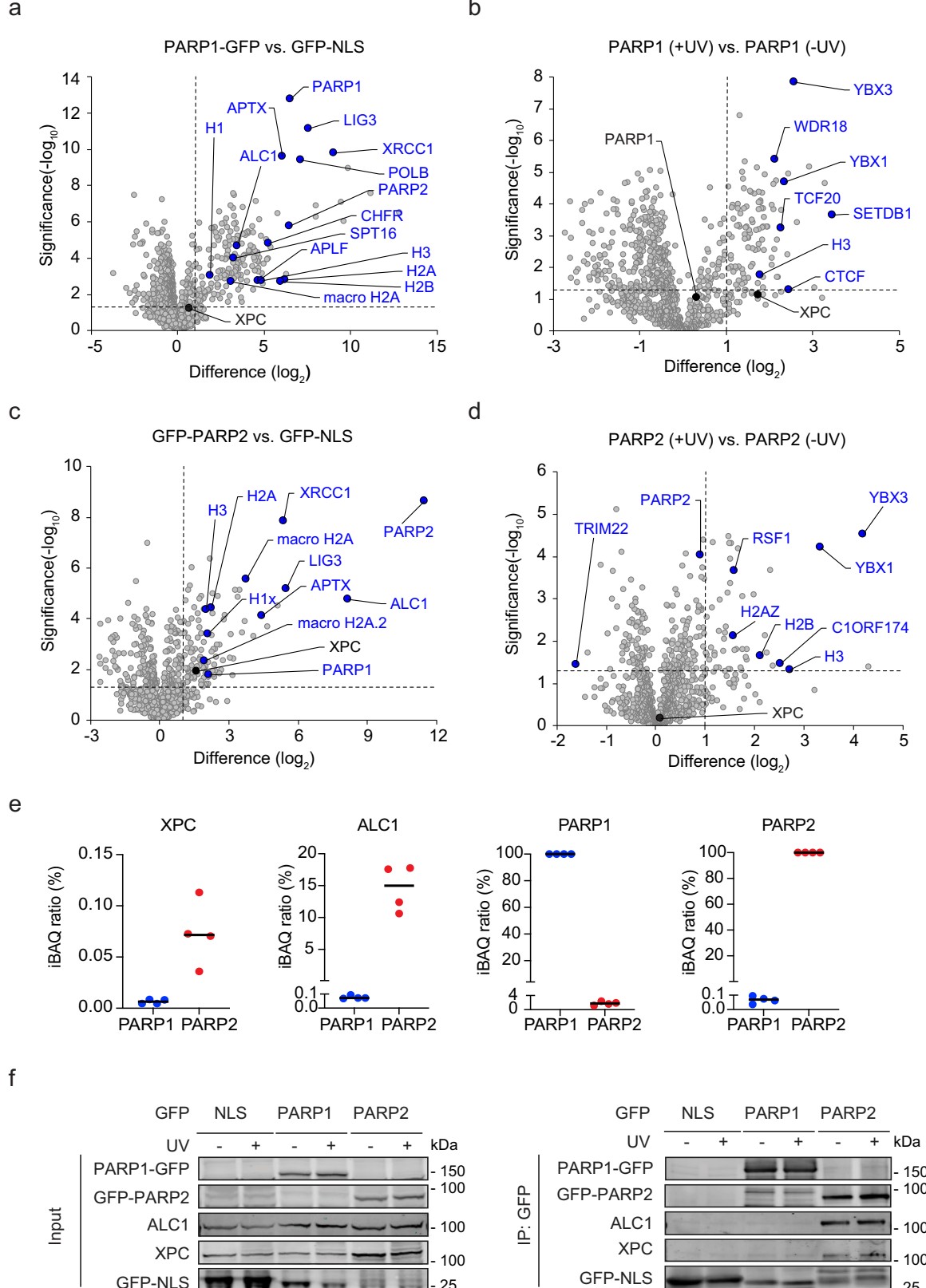

rapid recruitment of PARP1-GFP to sites of DNA damage within seconds, after which steady-state bound levels decreased in the first several minutes (Fig. 3a, b). GFP-PARP2 was also recruited to UV-C laser damage. Steady-state bound levels remained high for ~10 min without an apparent decrease

(Fig. 3c, d). The enrichment of PARP1-GFP shortly after UV-C micro-irradiation was more pronounced (1.4-fold) compared to the more modest recruitment of PARP2-GFP (1.2-fold). Interestingly, the recruitment kinetics of PARP1 and PARP2 were identical in XPC-KO cells (Fig. 3b, d). This indicates that

**Fig. 2 XPC and ALC1 interact more strongly with PARP2 than PARP1. a, c** Volcano plots displaying the interactomes of **a** PARP1-GFP and **c** GFP-PARP2 after GFP-pull-down from U2OS (FRT) WT cells and analysis by label-free proteomics. **b, d** Differential interactomes of **b** PARP1-GFP and **d** GFP-PARP2 from UV-C-irradiated (20 J/m², 1 h) vs. unirradiated U2OS (FRT) WT cells. **a–d** The dashed lines indicate a twofold enrichment on the x axis ($\log_2$ of 1) and a significance of 0.05 ($-\log_{10}$ P value of 1.3; two-sided t test) on the y axis. **e** Enrichment of XPC, ALC1, PARP1, and PARP2 in the co-IPs of PARP1 (**a**) and PARP2 (**b**), calculated by intensity-based quantification (IBAQ). Each data point represents a biological replicate (n = 4). **f** Co-IP of GFP-NLS, PARP1-GFP, and GFP-PARP2 in the presence and absence of UV-C (20 J/m², 1 h). Three independent replicates of each IP experiment were performed obtaining similar results.

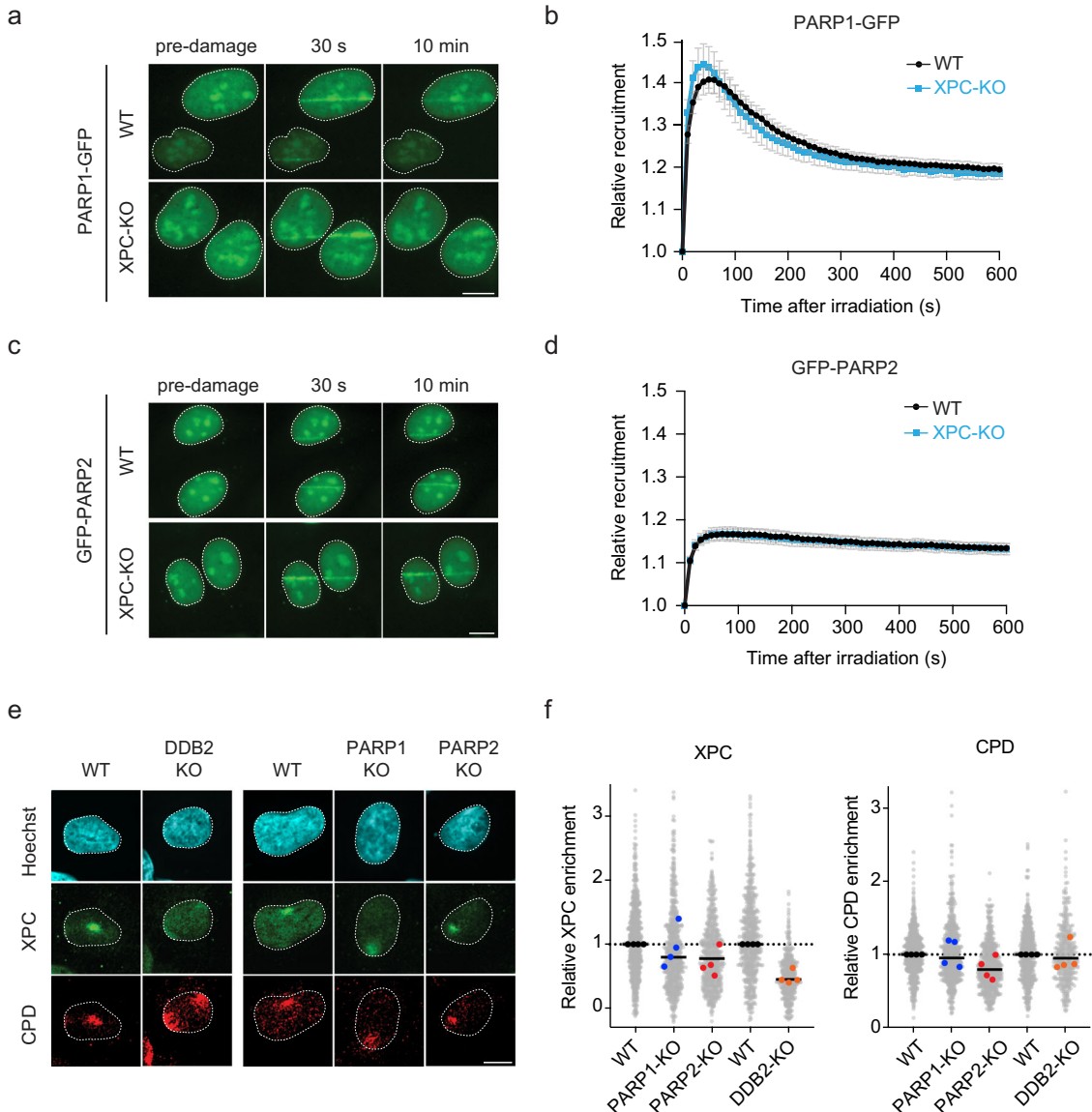

**Fig. 3 PARP1/2 recruitment to UV lesions is independent of XPC. a, c** Representative images of **a** PARP1-GFP and **c** GFP-PARP2 association with sites of local UV-C laser irradiation in U2OS (FRT) WT and XPC-KO cells at 30 sec and 10 min post-irradiation. **b, d** Kinetics of the recruitment of **b** PARP1-GFP and **d** GFP-PARP2 to and dissociation from UV-C lesions measured over 10 min in U2OS (FRT) WT and XPC-KO cells; 70–91 nuclei were analyzed in three independent biological replicates. The data are shown as mean + SEM normalized to pre-damage GFP intensity at micro-irradiation sites. **e** Representative images and **f** quantification of XPC colocalization with local UV-C irradiation sites (100 J/m²) marked by CPD, measured 10 min post-irradiation. In all, 58–230 cells were analyzed per condition. All cells are depicted as individual data points (gray). The medians of four biological replicates are depicted as colored points, while the bar represents the median of all data points. The scale bar in **a, c, e** is 5 μm.

PARP enzymes accumulate independently of XPC at sites of UV-induced DNA damage, which is in line with earlier findings showing that PARP1 recruitment is similar between WT and XPC-deficient cells at 10 min after UV irradiation[24].

Deletion of either PARP1 or PARP2 had a minor impact on the recruitment of XPC to UV lesions, as measured by immunofluorescence after local UV irradiation through micropore filters (Fig. 3e, f). The impact of PARP enzymes was milder than KO of DDB2, which led to considerably reduced XPC recruitment, as reported previously[16,20,39]. Our data thus suggest that XPC and PARP enzymes are recruited to DNA lesions largely independently, although PARP enzymes may stimulate XPC recruitment under certain conditions, such as low doses of DNA damage[20,24].

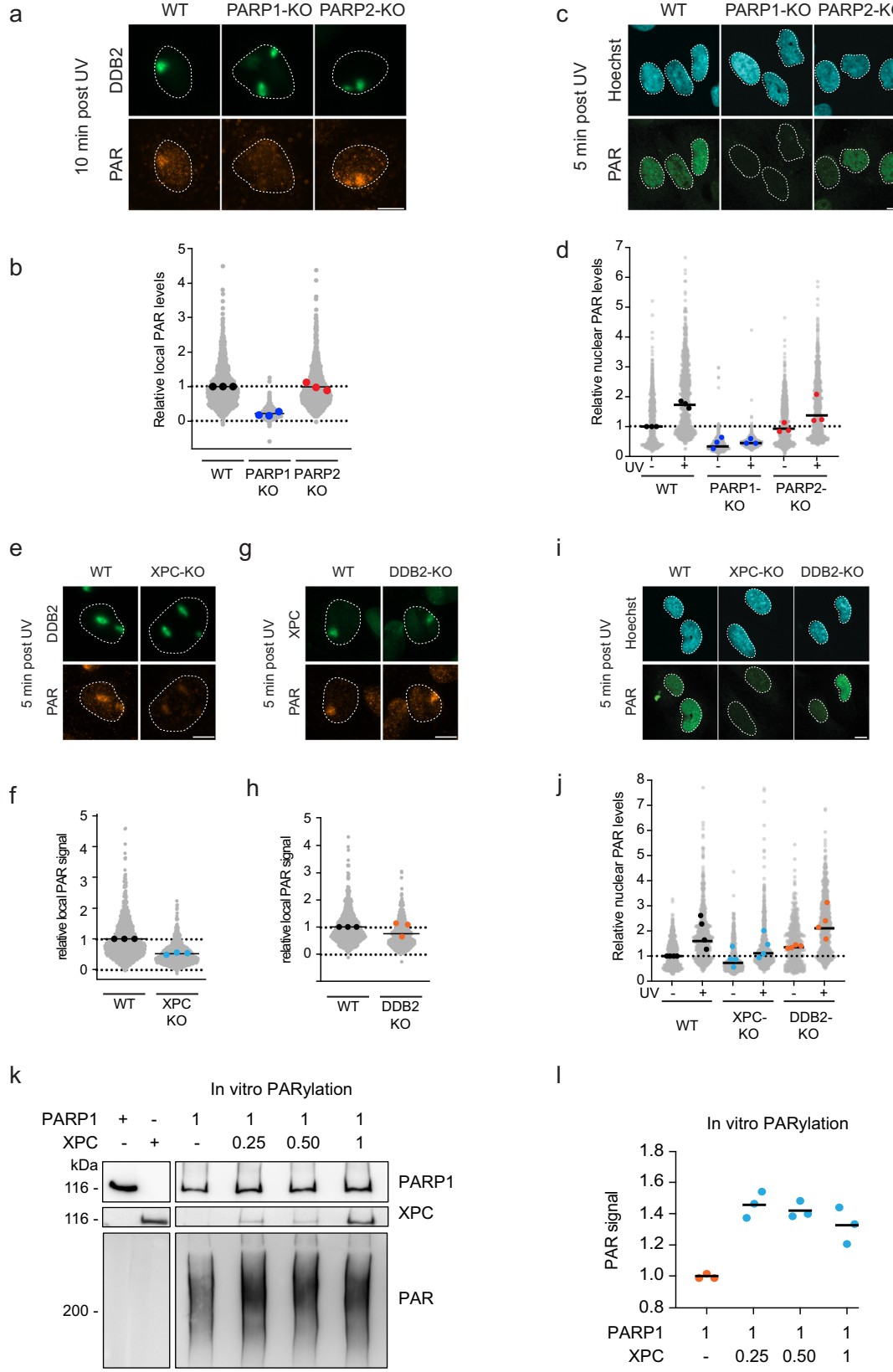

**XPC stimulates the poly-(ADP-ribose) response to UV lesions**. The PARP response involves the rapid and robust DNA damage-induced PARylation mainly of PARP1 itself, but also of other chromatin substrates[40]. To better understand how the PARP response modulates GGR, we measured nuclear PAR levels at sites of local UV-induced DNA damage by immunofluorescence

in PARP1-KO and PARP2-KO cells (Fig. 4a, b). PARylation strongly increased at sites of local UV irradiation, marked by DDB2 recruitment, in WT cells and PARP2-KO cells, but not in PARP1-KO cells (Figs. 4a, b; Supplementary Fig. 2a). Similarly, nuclear PAR levels also increased after global UV irradiation in a manner that was dependent on PARP1, but not on PARP2

**Fig. 4 XPC stimulates the poly-(ADP-ribose) response at UV lesions. a** Representative images and **b** quantification of poly-(ADP-ribose) (PAR) levels 10 minutes after local UV-C irradiation (30 J/m$^2$) by immunofluorescence (Trevigen, 4335-MC-100) in the indicated cells. Quantification of DDB2 levels is shown in Fig. S2a. >100 cells were analyzed per condition from three independent experiments. **c** Representative images and **d** quantification of poly-(ADP-ribose) (PAR) levels 5 minutes after UV-C irradiation (20 J/m$^2$) by immunofluorescence (Millipore; MABE1031) in the indicated cells. >75 cells were analyzed per condition from three independent experiments. Additional representative images are found in Fig. S2b. **e** Representative images and **f** quantification of poly-(ADP-ribose) (PAR) levels 10 minutes after local UV-C irradiation (30 J/m$^2$) by immunofluorescence (Trevigen, 4335-MC-100) in in the indicated cells. Quantification of DDB2 levels is shown in Fig. S2c. >100 cells were analyzed per condition from 3 independent experiments. **g** Representative images and **h** quantification of poly-(ADP-ribose) (PAR) levels 10 minutes after local UV-C irradiation (30 J/m$^2$) by immunofluorescence (Trevigen, 4335-MC-100) in the indicated cells. Quantification of XPC levels is shown in Fig. S2d. >100 cells were analyzed per condition from 3 independent experiments. **i** Representative images and **j** quantification of poly-(ADP-ribose) (PAR) levels 5 minutes after UV-C irradiation (20 J/m$^2$) by immunofluorescence (Millipore; MABE1031) in the indicated cells. >65 cells were analyzed per condition from four independent experiments. **b, d, f, h, j** All cells are depicted as individual data points (gray). The median of each biological replicate is depicted as a colored point, while the bar represents the median of all data points. **k** Representative images and **l** quantification of the PARylation assay in which recombinant PARP1 (1 pmol) together with NAD was incubated with recombinant XPC-RAD23B (0, 0.25, 0.5, and 1 pmol) for 5 min after which UV-irradiated plasmid was added to the mixture for 30 min. The reaction was stopped and PARylation of PARP1 was monitored. The colored points represent the individual quantification from three independent experiments. The bar represents the median of all data points. The scale bar in **a, c, e, g, i** is 5 μm.

(Fig. 4c, d; Supplementary Fig. 2b). This demonstrates that the PAR response at UV lesions is largely dependent on PARP1, as demonstrated for other DNA-damaging agents[29,41–43].

Having established that XPC interacts with PARP1 and PARP2 (Figs. 1 and 2), we sought to investigate whether XPC impacts the PAR response. Strikingly, XPC-KO cells showed an attenuated PAR response at sites of local UV damage, while DDB2-KO cells established PAR levels similar to wild-type U2OS cells (Fig. 4e–h; Supplementary Fig. 2c, d). Identical results were obtained when PAR levels were monitored after global UV irradiation (Figs. 4i, j; Supplementary Fig. 2e). Knockdown of XPA in XPC-deficient cells did not further reduce DNA damage-induced PARylation at sites of local UV-induced DNA damage (Supplementary Fig. 2f±i), suggesting that this phenomenon is dependent on GGR and not on transcription-coupled repair. To understand whether XPC may directly stimulate the activity of PARP enzymes through their interaction, we performed in vitro PARylation assays using recombinant PARP1 in the presence of UV-irradiated DNA and increasing amounts of recombinant XPC-RAD23B complex (Fig. 4k, l). In this minimal in vitro system without additional components, we observed that XPC directly stimulated the catalytic activity of PARP1 by ~1.5 fold, which was already observed at a 4:1 ratio of PARP1 over XPC-RAD23B complex and did not increase when more XPC was added to the reaction (Fig. 4k, l). These findings show that XPC stimulates the initial and rapid PAR response at UV lesions by enhancing the protein activity of PARP1.

**PARP1 and ALC1 are UV-induced substrates of PARylation.** Based on the close interaction of XPC and PARP1/2, we next asked which of these proteins become PARylated upon UV damage. To this end, we performed pull-down experiments under high-salt conditions to disrupt protein-protein interactions and capture the PARylation status of the immunoprecipitated proteins. Pull-down of PARP1-GFP revealed robust PARylation in response to UV irradiation (Fig. 5a). By contrast, GFP-PARP2 and XPC-GFP were already PARylated in control cells at lower levels with no further increase following UV irradiation (Fig. 5b, c). The poly-(ADP-ribose)-dependent chromatin remodeler ALC1 was also strongly PARylated in response to UV irradiation (Fig. 5d).

To better capture the dynamics of protein PARylation in response to UV irradiation, we performed an adapted LacO-based colocalization assay, in which we fused the PAR-binding macrodomain of macroH2A1.1 to LacR and tethered this PAR-binding module to a LacO array in U2OS 263 cells[44,45]. We then induced the PARylation response by local irradiation with a

266 nm UV-C laser. Micro-irradiation triggered the recruitment of GFP-tagged versions of XPC, ALC1, PARP1, and PARP2 to sites of UV-C-induced laser damage (Figs. 5e, f and Supplementary Fig. 3a). Within one minute following irradiation, we also detected the capture of XPC, ALC1 and PARP1 at the LacO array bound by the LacR-fused macrodomain PAR-binding module. This suggests that protein complexes containing XPC, ALC1 and PARP1 become PARylated at laser micro-irradiation sites and dissociate from these sites in a modified, PARylated state, which in turn allows their interaction with the immobilized macrodomain at the LacO array. Treatment with the PARP inhibitor olaparib prevented the accumulation of XPC at the LacO site, demonstrating that its capture at the LacO array is fully dependent on PARylation (Fig. 5e, f). Interestingly, PARP2 was recruited to UV sites, but could not be detected at the LacO site following micro-irradiation, suggesting that its UV-induced PARylation is not sufficiently high or that the protein is not sufficiently mobile to enable capture of PARP2 by the immobilized PAR-binding module at the LacO array.

The capture of either PARP1 or ALC1 at the immobilized PAR-binding module is consistent with their UV-induced PARylation detected in pull-down experiments (Fig. 5a, d). In contrast, XPC is captured by the immobilized PAR-binding module, but we could not detect UV-induced PARylation of XPC following immunoprecipitation (Fig. 5c). This means that either the capture of XPC at the immobilized PAR-binding module is mediated indirectly through its interaction with PARP1, which is heavily PARylated after UV (Fig. 5a). Alternatively, XPC may undergo a conformational change at sites of DNA damage that results in the exposure of its PARylated residues and its UV-induced capture at the immobilized PAR-binding module. Either way, our data indicate that the PARP1-XPC complex and chromatin remodeler ALC1 become robustly PARylated at sites of UV-induced DNA damage.

**PARP1 and PARP2 protect against UV-induced DNA damage.** To establish the relevance of PARP1 and PARP2 in UV damage repair, we first assayed PARP1-KO and PARP2-KO cells for UV sensitivity. Western blot analysis confirmed the knockout of PARP1 or PARP2 using specific antibodies (Fig. 6a). Clonogenic survival assays showed that KO of either PARP1 or PARP2 conferred sensitivity to UV irradiation in human cells (Fig. 6b). The siRNA-mediated knockdown of XPC, which was confirmed by western blot analysis, caused similar sensitivities in wild-type and PARP-deficient cells, suggesting that the PARP enzymes cause UV sensitivity mainly through the GGR pathway (Supplementary Fig. 3b, c). To further validate these findings, we

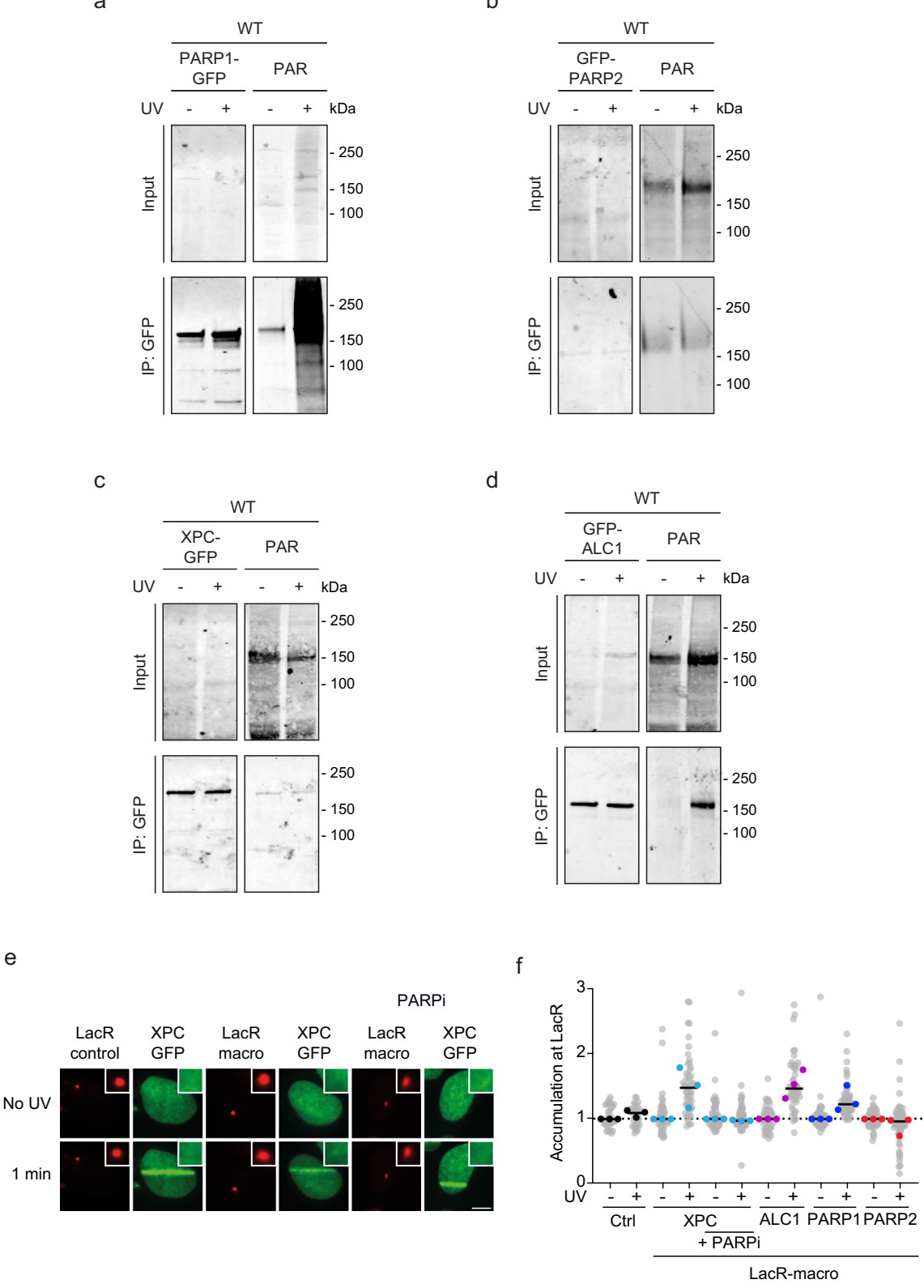

**Fig. 5 PARP1 and ALC1 are PARylated in response to UV. a–d** Immunoprecipitation of **a** PARP1-GFP, **b** GFP-PARP2, **c** XPC-GFP, **d** GFP-ALC1 under high-salt conditions in the presence and absence of UV-C (20 J/m², 15 min) stained for PAR (Millipore; MABE1016) or GFP. Three independent replicates of each IP experiment were performed obtaining similar results. **e** Representative images and **f** quantification of GFP-tagged XPC, ALC1, PARP1, or PARP2 recruitment to the LacO array upon tethering to the indicated mCherry-LacR-macrodomain. Pictures were taken before and 1 min after UV-C micro-irradiation. 24–45 nuclei were analyzed in three independent biological replicates (*n* = 3). Additional representative images are found in Fig. S3a. The scale bar in **e** is 5 μm.

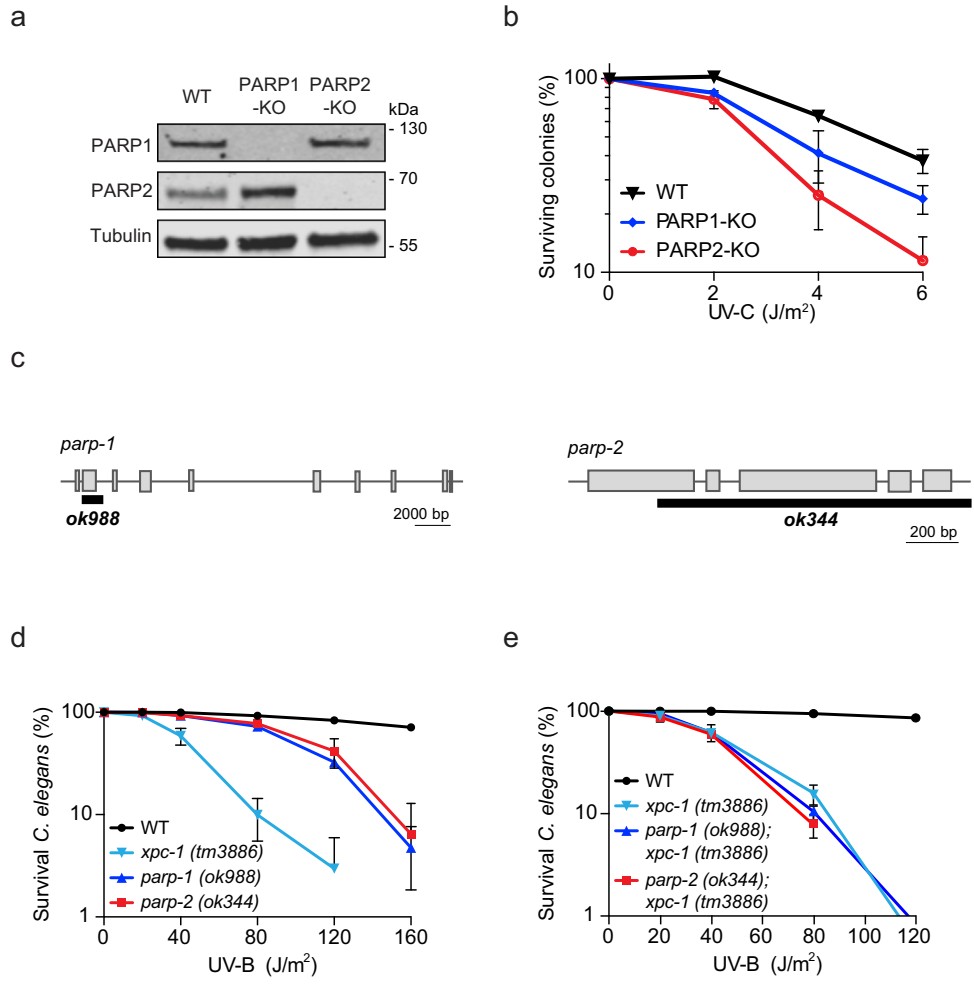

**Fig. 6 PARP1 and PARP2 impact the repair of CPD lesions. a** Western blot of U2OS WT, PARP1-KO and PARP2-KO cells. **b** Clonogenic survival assays of U2OS WT, PARP1-KO, and PARP2-KO cells upon UV-C irradiation. The data is depicted as mean + S.E.M. from $n = 4$ independent experiments, except for PARP1-KO for which $n = 3$. **c** Representation of the deletions in the *C. elegans* parp-1 (ok988) and parp-2 (ok344) strains. **d** Germ cell and embryo UV survival assays of PARP1-deficient parp-1(ok988) and PARP2-deficient parp-2(ok344) *C. elegans*. **e** Germ cell and embryo UV survival assays of XPC-deficient xpc-1 (tm3886), XPC and PARP1 double deficient (ok988; tm3886), or XPC and PARP2 double deficient (ok344; tm3886) *C. elegans*. **d**, **e** The data are depicted as mean + S.E.M. from three independent experiments.

performed CPD dot blot assays in PARP1-KO and PARP2-KO cells. To this end, genomic DNA was isolated from cells at varying time points after UV-C irradiation to determine the amount of remaining UV-induced photoproduct. This approach revealed that both PARP1-KO and PARP2-KO cells displayed delayed CPD repair (Supplementary Fig. 3d, e).

To extend the biological relevance of these findings, we asked whether the role of PARP1 and PARP2 in protecting against UV irradiation is evolutionarily conserved in an animal model. We obtained PARP1-deficient (ok988) and PARP2-deficient (ok344) *C. elegans* (Fig. 6c) and performed germ cell and embryo survival assays after UV irradiation, which specifically monitor GGR[46]. Deletion of either PARP1 or PARP2 strongly sensitized nematodes to UV-B light compared to WT animals (Fig. 6d). Interestingly, animals containing a double knockout for XPC (tm3886) and either PARP1 (ok988) or PARP2 (ok344), were as sensitive to UV-B irradiation as single XPC-deficient nematodes (Fig. 6e). These findings indicate evolutionary conservation of the involvement of both PARP enzymes in GGR.

**ALC1 is recruited to UV lesions by XPC**. Our proteomics analyses revealed that XPC binds more robustly to PARP2 than PARP1, and that ALC1 is an abundant interactor of PARP2

(Fig. 2). Moreover, ALC1 was robustly PARylated in response to UV irradiation (Fig. 5d). To better understand the links between XPC, PARP enzymes, poly-(ADP-ribose)-responses and ALC1 in GGR, we profiled the ALC1 interactome, dissected the role of its PAR-binding macrodomain and ATPase activity, as well as measured the UV-C-dependent recruitment dynamics of ALC1. We, therefore, generated ALC1-KO cells in U2OS (FRT) and stably re-expressed GFP-tagged versions of ALC1 (WT, ATPase-dead; E175Q, and PAR-binding-deficient; Δmacrodomain; Supplementary Fig. 4a). Label-free proteomics after GFP-ALC1 pull-down confirmed a strong interaction of wild-type ALC1 with PARP2, PARP1, the FACT subunit SPT16 and core histones (Fig. 7a). In contrast, the ATPase-dead version of ALC1 interacted less with PARP2, and core histones, suggesting that the ATPase activity of ALC1 impacts the association of the enzyme with PARP2 (Fig. 7b). Consistently, immunoprecipitation experiments confirmed that ALC1 robustly bound PARP2 and to a lesser extent PARP1, and that these interactions were decreased with ALC1 E175Q (Fig. 7c). Moreover, an ALC1 Δmacrodomain mutant showed a completely disrupted interaction with PARP1/2.

Next, we tested the recruitment of ALC1 to UV-C DNA damage sites. Local UV-C laser irradiation experiments showed

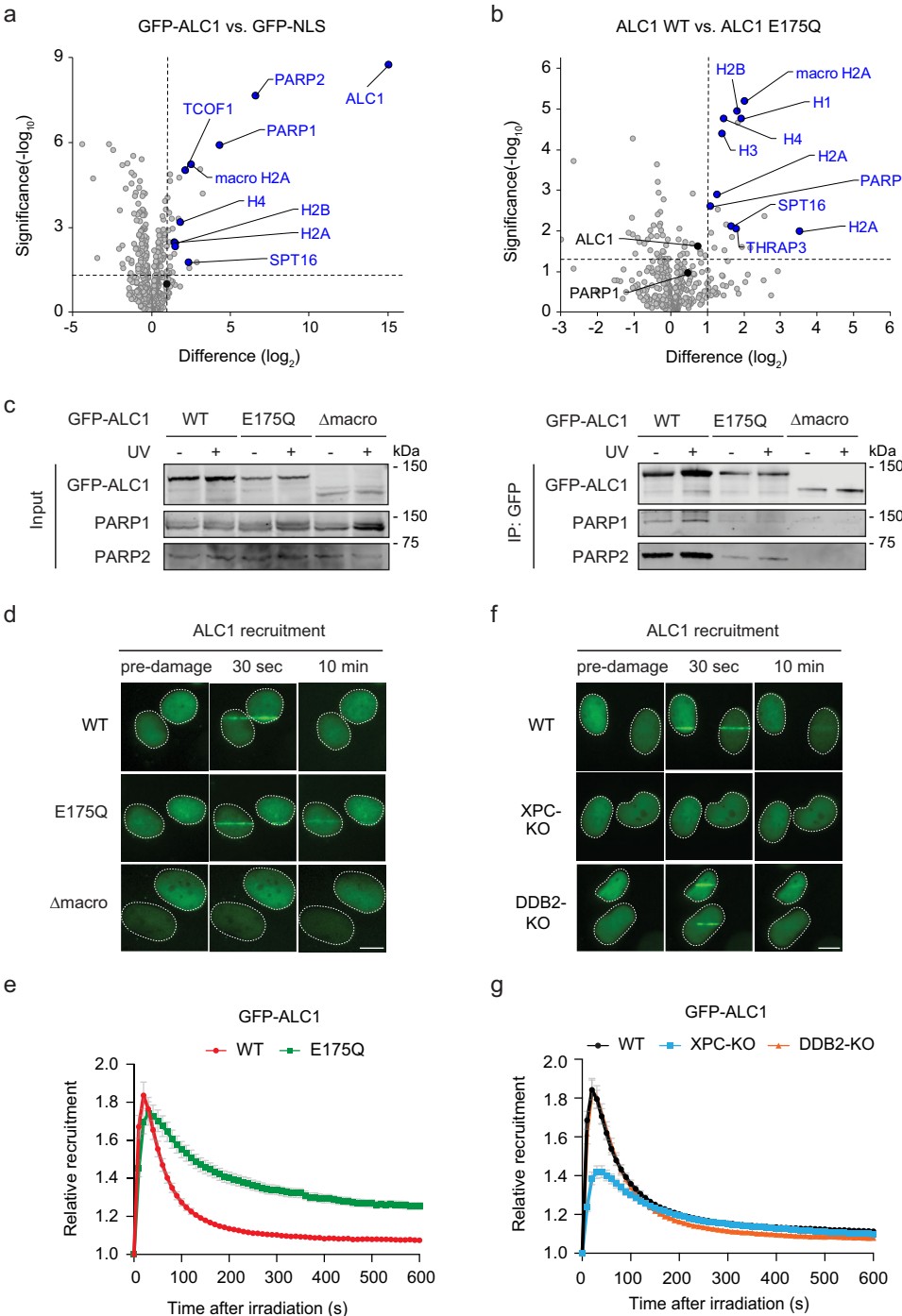

**Fig. 7 The chromatin remodeler ALC1 is recruited to UV lesions by XPC. a** Volcano plot displaying the interactomes of GFP-ALC1 over GFP-NS after GFP-pull-down from U2OS (FRT) ALC1-KO cells and analysis by label-free proteomics. **b** Differential interactome of GFP-ALC1 WT vs. the catalytic-deficient GFP-ALC1 E175Q mutant. **a**, **b** The dashed lines indicate a twofold enrichment on the x axis ($\log_2$ of 1) and a significance of 0.05 ($-\log_{10}$ P value of 1.3; two-sided t test) on the y axis. **c** Co-IP of GFP-ALC1 WT, GFP-ALC1 E175Q, and a PAR-binding-deficient GFP-ALC1 Δmacrodomain in the presence and absence of UV-C (20 J/m², 1 h). Three independent replicates of each IP experiment were performed obtaining similar results. **d** Representative images and **e** recruitment kinetics of GFP-ALC1 WT, GFP-ALC1 E175Q, GFP-ALC1 Δmacrodomain at sites of local UV-C laser irradiation in U2OS (FRT) ALC1-KO cells. 77–82 cells were analyzed in three independent experiments. **f** Representative images and **g** recruitment kinetics of GFP-ALC1 at sites of local UV-C laser irradiation in U2OS (FRT) WT, XPC-KO, and DDB2-KO cells. 107–135 cells were analyzed in n = 4 (WT), n = 5 (XPC-KO) or n = 6 (DDB2-KO) independent experiments. The data are shown as mean + SEM normalized to pre-damage GFP intensity at micro-irradiation sites. The scale bar in **d**, **f** is 5 μm.

that GFP-ALC1 was rapidly recruited with similar kinetics as PARP1 independently of its ATPase activity, as the WT and E175Q versions of ALC1 showed similar association kinetics (Fig. 7d, e). However, the ATPase mutant remained at UV-damaged sites for longer times. The macrodomain deletion mutant failed to recruit (Fig. 7d, e). Our data suggest that the chromatin remodeler ALC1 is tightly linked to the poly-(ADP-ribose) response to UV lesions. The PAR-dependent recruitment of ALC1 via its macrodomain leads to a robust interaction with chromatin, which is then followed by ATP hydrolysis and localized chromatin remodeling. In turn, ATP hydrolysis by ALC1 results in the displacement and dissociation of the ALC1 chromatin remodeler from UV-damaged chromatin.

Further, the recruitment of ALC was fully impaired in PARP1-deficient cells, while its recruitment still occurred in PARP2-deficient cells, albeit with distinct kinetics (Supplementary Fig. 4b, c). This suggests that PARP2 may modulate ALC1 association dynamics. Strikingly, we also observed a strongly decreased recruitment of ALC1 in XPC-KO cells, while the recruitment of ALC1 in DDB2-KO cells was similar to WT (Fig. 7f, g). Our data indicate that the GGR-initiator protein XPC stimulates the recruitment of chromatin remodeler ALC1 to sites of UV-induced DNA damage, suggesting that the XPC-mediated stimulation of the poly-(ADP-ribose) response promotes PAR-dependent downstream processes, such as ALC1-mediated chromatin remodeling.

**ALC1 stimulates the clearing of genomic UV lesions**. To directly assess the relevance of ALC1 in UV damage repair, we performed slot blot assays to monitor 6-4PP and CPD levels in ALC1-KO cells. Deletion of the chromatin remodeler ALC1 resulted in a delayed repair of 6-4PPs (Fig. 8a; S5a) and a strong repair defect of CPD lesions (Fig. 8b; Supplementary 5b) compared to XPC-KO cells. Interestingly, the deletion of ALC1 did not affect the transcription-coupled sub-pathway of NER, as ALC1-KO cells were not sensitive to the drug Illudin S in clonogenic survival assays and did not impact the recovery of RNA synthesis after UV damage, as measured by 5-EU incorporation (Supplementary Fig. 5c–e). Our data suggest that ALC1 specifically acts in GGR.

**The ATPase activity of ALC1 stimulates XPC-dependent DNA repair**. Having found that XPC stimulates the recruitment of ALC1 to repair sites, we wanted to understand how ALC1-mediated chromatin remodeling affects UV damage repair. Loss ALC1 did not affect the recruitment of XPC or DDB2 following UV-C laser irradiation (Supplementary Fig. 6a, b), suggesting that XPC and DDB2 binding precedes ALC1 recruitment and that ALC1 does not affect the lesion-recognition step of GGR. Importantly, clonogenic UV survival experiments revealed that the knockout of ALC1 rendered cells sensitive to UV irradiation to a similar extent as DDB2-KO cells (Fig. 8c). Re-expression of ALC1 WT, but not the ATPase-dead or PAR-binding-deficient ALC1 mutants, rescued both UV-sensitivity phenotype (Fig. 8d) and unscheduled DNA synthesis at UV sites of ALC1-KO cells (Fig. 8e, f). These data indicate that active, ALC1-catalyzed chromatin remodeling plays an important role in UV repair.

To directly monitor UV-induced chromatin changes induced by PARP1/2 and the ALC1 chromatin remodeler, we sequentially irradiated cells expressing photoactivatable GFP fused to histone H2A (PAGFP-H2A) with a UV-C laser (266 nm) to generate UV-specific photolesions, immediately followed by UV-A laser (355 nm) irradiation to activate PAGFP-H2A specifically at sites of local UV damage (Fig. 9a). Control experiments showed that UV-A laser irradiation alone activated PAGFP-H2A, but failed to recruit the GGR protein DDB2-mCherry, whereas sequential irradiation with UV-C and UV-A lasers triggered DDB2-mCherry recruitment and locally activated PAGFP-H2A (Supplementary Fig. 6c). Importantly, combined irradiation with UV-C and UV-A lasers failed to recruit the double-strand break repair protein NBS1-mCherry at sites that did show local activation of PAGFP-H2A (Supplementary Fig. 6d). This shows that UV-C/UV-A laser irradiation without BrdU sensitization does not cause the substantial formation of double-strand DNA breaks, enabling us to specifically capture chromatin changes at UV-C-induced DNA lesions.

While wild-type and PARP2-KO cells showed considerable expansion of PAGFP-H2A tracks, indicative of DNA damage-induced chromatin remodeling, following sequential UV-C and UV-A laser irradiation, such an expansion was attenuated in either PARP1-KO or ALC1-KO cells. These findings reveal that chromatin expansion at sites of UV-C-induced DNA damage is stimulated by PARP1-dependent and ALC1-mediated chromatin remodeling (Fig. 9a, b).

**Loss of ALC1 catalytic activity leads to a hyper-PAR response at UV lesions**. Our previous work revealed that cells mount a hyper-PAR response to single-stranded DNA breaks in the absence of ALC1, resulting in the trapping of PARP2 at these structures[47]. Having shown that ALC1's catalytic activity is required for chromatin expansion at sites of UV lesions and efficient GGR, we wondered whether the loss of ALC1 could also affect the PAR response to UV lesions.

To monitor the PAR response, we locally irradiated cells with UV-C light and fixed cells at different time points after irradiation. Wild-type cells showed a clear PAR signal at sites of UV-induced lesions, marked by the local enrichment of XPC (Supplementary Fig. 6e), which was similar between all time points examined (Fig. 9c, d). In contrast, ALC1-KO cells initially mounted a similar PAR response shortly after UV irradiation, but PAR levels steadily increased over time to ~2-fold higher levels at 30 min compared to wild-type cells (Fig. 9c, d). This hyper-PAR response could be fully rescued by expression of ALC1$^{WT}$, while expression of ATPase inactive ALC1 E175Q even further increased PAR levels to ~3-fold over wild-type cells (Fig. 9c, d). These findings show that the catalytic activity of ALC1 is required to shut off the PAR response at UV lesions. Altogether, these findings demonstrate that both the recruitment of the chromatin remodeler ALC1 to PAR chains and its ability to function as an ATP-driven chromatin remodeler is critical for UV repair.

## Discussion

The UV lesion-recognition factor XPC initiates repair of helix-destabilizing DNA lesions, but is less efficient to initiate repair of lesions that cause poor helix destabilization. Here, we identify a biochemical complex of XPC with both PARP1 and PARP2. The XPC-PARP1 complex is key to the PAR-mediated recruitment of the chromatin remodeler ALC1. In turn, ATP-catalyzed chromatin remodeling by the ALC1 helicase powers and promotes efficient CPD repair. The mechanistic role of PARP2 is less clear, although evolutionarily conserved from human cells to nematodes. Our work identifies a novel XPC-PARP axis that links ALC1-mediated chromatin remodeling to global genome nucleotide excision repair (Fig. 9e).

Proteomic analyses identify PARP1 and PARP2 as intricate interactors of the lesion-recognition protein XPC. Deletion of XPC resulted in a reduced PAR response at UV-C lesions and impaired the efficient recruitment of the PAR-dependent chromatin remodeler ALC1. Our work suggests that XPC acts as a key regulator of PARylation in GGR, likely by stimulating the

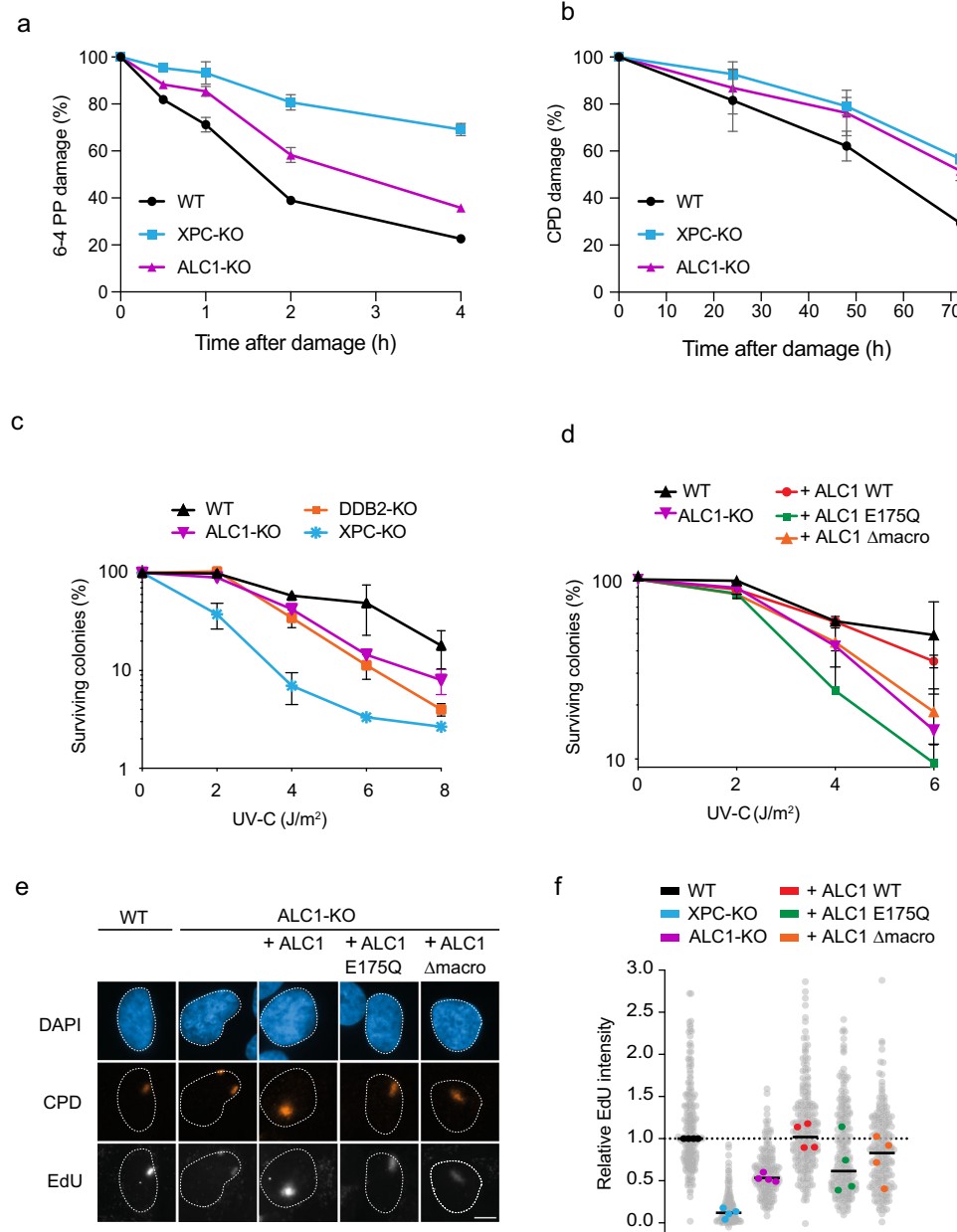

**Fig. 8 ALC1's ATPase activity stimulates XPC-dependent DNA repair. a** Quantification of 6-4PP levels based on slotblot in U2OS (FRT) WT, XPC-KO, and ALC1-KO cells at different time points after UV-C damage (20 J/m²). The data is depicted as mean + S.E.M. of four independent experiments. Representative dot blots are shown in Fig. S5a. **b** Quantification of CPD levels based on slotblot in U2OS (FRT) WT, XPC-KO, and ALC1-KO cells at different time points after UV-C damage (20 J/m²). The data is depicted as mean + S.E.M. of four independent experiments where each experiment is based on two technical replicates. Representative dot blots are shown in Fig. S5a. **c, d** Clonogenic survival assays of **c** U2OS (FRT) WT, ALC1-KO, DDB2-KO, and XPC-KO cells as well as **d** U2OS (FRT) WT, ALC1-KO, ALC1-KO + GFP-ALC1, ALC1-KO + GFP-ALC1 E175Q, ALC1-KO + GFP-ALC1 Δmacrodomain cells upon UV-C irradiation. The data are depicted as mean + S.E.M. from three independent experiments. **e** Representative images and **f** quantification of unscheduled DNA synthesis experiments in U2OS (FRT) WT, XPC-KO, ALC1-KO, ALC1-KO + GFP-ALC1, ALC1-KO + GFP-ALC1 E175Q, ALC1-KO + GFP-ALC1 Δmacrodomain cells upon UV-C irradiation. >39 cells were analyzed in four independent experiments. All cells are depicted as individual data points (gray). The median of each biological replicate is depicted as a colored point, whereas the bar represents the median of all data points. The scale bar in **e** is 5 μm.

enzymatic activity of PARP1. Although several studies identified XPC as a target of PARylation in vitro and upon exogenous DNA damage treatment of cultured cells with $H_2O_2$[48–50], our analysis shows that XPC is not PARylated in response to UV-C irradiation. By contrast, PARP1 and ALC1 become extensively decorated with PAR in UV-irradiated cells. Intriguingly, after dissociation from UV lesions, XPC was captured by an immobilized PAR-binding module in the same cell nucleus, even

though we could not detect increased PARylation of XPC beyond its basal steady-state PARylation in pull-down experiments. We envision that the robust UV-induced PARylation of PARP1 may indirectly recruit XPC to the immobilized PAR-binding module. Alternatively, XPC may undergo a conformational change at damaged sites, resulting in the exposure of its steady-state PARylated residues, leading to its capture at the immobilized PAR-binding module. Deducing the mechanism of PAR

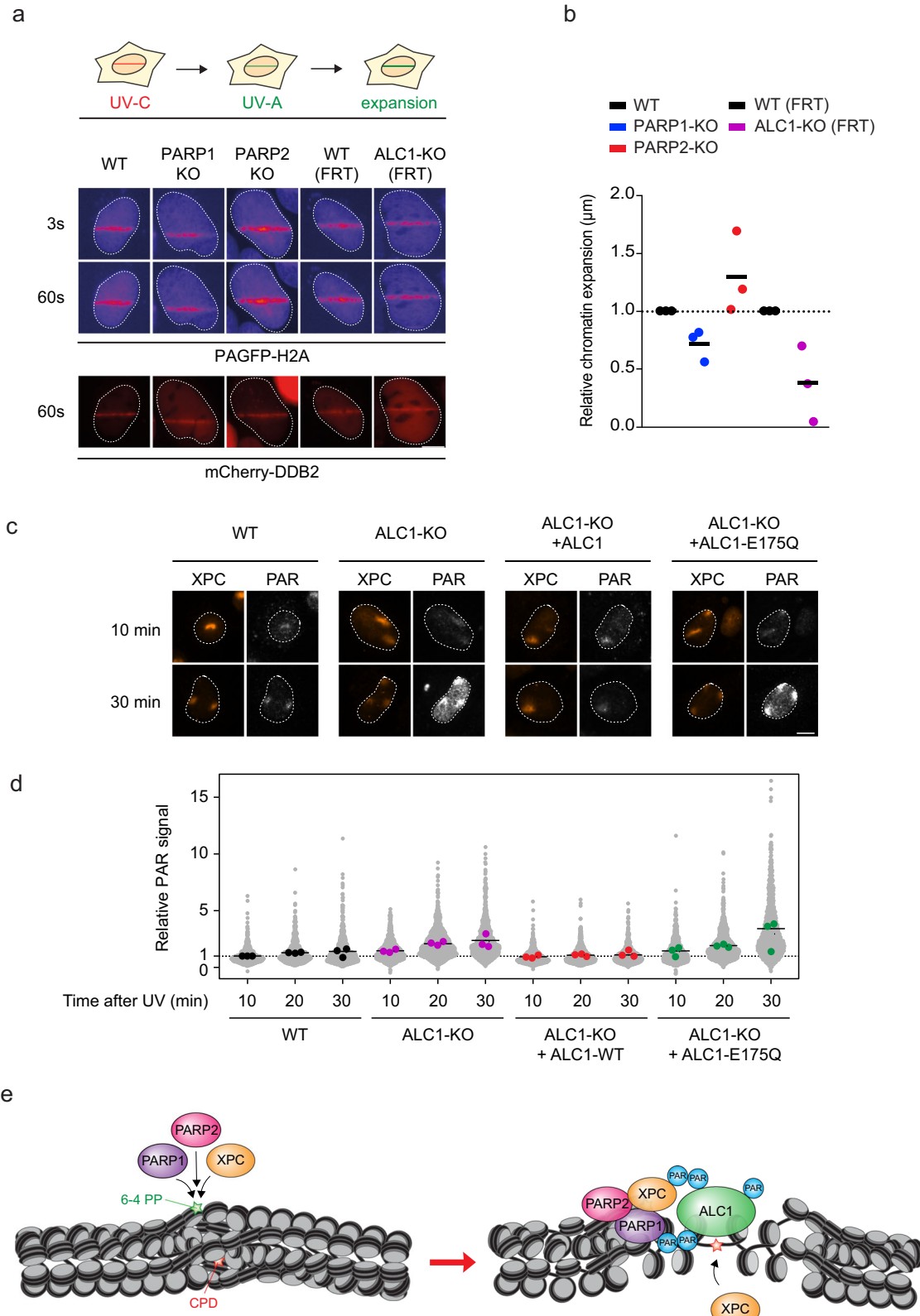

regulation by XPC may help us further understand the impact of PARylation on the protein networks involved in GGR.

In several DNA repair pathways, PARP1 is accompanied by a second poly-(ADP-ribose) polymerase, PARP2. The coordinated action of both PARP1 and PARP2 seems to be required for efficient base excision repair and DNA double-strand break repair[34,51,52]. However, the precise function of PARP2 has so far largely remained elusive and has not yet been described in nucleotide excision repair. Here, we identify PARP2 as a novel regulator of the GGR response. The protein displays abundant interactions with XPC and especially with the chromatin remodeler ALC1, suggesting that it is tightly linked to the newly

**Fig. 9 ALC1 regulates UV-induced chromatin expansion and PAR response shut-down. a** Outline of the sequential UV-C and UV-A laser irradiation approach and representative images of PAGFP-H2A and mCherry-DDB2 in the indicated cell lines at 3 s and 60 s after sequential irradiation. Additional irradiation controls are shown in Fig. S6c, d. **b** Quantification of the UV-induced expansion of PAGFP-H2A tracks marked by mCherry-DDB2 recruitment at 60 s after sequential irradiation. The normalized expansion of each biological replicate is depicted as a point, while the bar represents the median from three independent experiments. The expansion in each experiment for PARP1-KO (blue dots) and PARP2-KO (red dots) was normalized to the isogenic WT control (black dots), while ALC1-KO (purple dots) was normalized to the isogenic WT (FRT) control (black dots). >38 cells were analyzed in 3 independent experiments. **c** Representative images and **d** quantification of poly-(ADP-ribose) (PAR) levels 10, 20, and 30 minutes after local UV-C irradiation (30 J/m$^2$) by immunofluorescence (Trevigen, 4335-MC-100) in U2OS WT, ALC1-KO, ALC1-KO + GFP-ALC1, ALC1-KO + GFP-ALC1 E175Q. Quantification of XPC levels in the same cells is shown in Fig. S6e. The median of each biological replicate is depicted as a colored point, while the bar represents the median of all data points. >80 cells were analyzed per condition from three independent experiments. The scale bar in **a**, **c** is 5 µm. **e** Model of PARP1, PARP2, and ALC1 influence on XPC-dependent repair.

identified XPC-PARP axis. Interestingly, while the contribution of PARP2 to the UV-induced PAR response was minor, we found that PARP2 deletion strongly sensitized cells to UV and was important for the repair of difficult-to-repair CPD lesions. This sparks the question of whether PARP2 may contribute to GGR independent of its catalytic activity. Such a mechanism of regulation was proposed previously for the efficient repair of DNA double-strand breaks by homologous recombination, where PARP2 limits 53BP1 accumulation and promotes end-resection independently of its catalytic activity[52]. Alternatively, PARP2 may contribute to the synthesis of distinct PAR chains, such as branched PAR molecules at UV-C lesions[32]. Smaller quantities of branched PAR chains may be sufficient to promote efficient GGR by virtue of their recognition by specific DNA repair factors, including potentially ALC1.

The newly identified XPC-PARP axis is tightly linked to ALC1-mediated chromatin remodeling in GGR. The deletion of either XPC, PARP1 or PARP2 abrogates, completely abolishes, or modulates the timely recruitment of ALC1 to UV-C lesions, respectively. This underlines the tight functional connection between the four proteins. Surprisingly, the recruitment of ALC1 was not effectively impaired by the deletion of DDB2, as previously described[28]. The DDB2-dependent recruitment observed previously was pronounced in the absence of XPA, while being mild in wild-type cells. Based on our recruitment data, we suggest that the XPC-PARP axis is the dominant route to recruit ALC1 to UV-C lesions. DDB2 may support XPC in recruiting ALC1 in situations of increased or sustained damage, as observed in the absence of XPA. This hypothesis is supported by earlier findings showing that inhibition of PARP enzymes strongly delays GGR-mediated repair also in DDB2-deficient cells[24].

Our work identifies a key role of active chromatin remodeling in GGR. ALC1 is an ATP-dependent SNF2-type chromatin remodeler that is activated by an enzymatic switch upon poly-(ADP-ribose) binding via its C-terminal macrodomain[37,38]. The fast activation of ALC1's chromatin remodeling activity results in the local opening of chromatin around DNA lesions[53]. Here, we established that both PAR-binding and active chromatin remodeling by ALC1 is required for efficient GGR. Moreover, both PARP1 and ALC1 stimulate chromatin expansion at UV lesions, lending direct support for ALC1-catalyzed chromatin remodeling during GGR. Interestingly, ALC1-KO cells did not reveal any defects in transcription-coupled repair, suggesting that the activity of ALC1 is specifically required for the GGR pathway. In addition, ALC1, together with the poly-(ADP-ribose) polymerases PARP1 and PARP2, show a strong preference for supporting the repair of CPD lesions, rather than of 6-4PPs. It should be noted that earlier studies also reported delayed 6-4PP repair upon depletion of PARP1 measured by flow cytometry[24,29].

Given the difficulty in recognizing and the increased time required for CPD repair, it is likely that this repair pathway requires additional chromatin factors to facilitate the repair of

such challenging lesions. While PARP1 and PARP2 could promote chromatin loosening through the establishment of negatively charged PAR chains on chromatin components around the DNA lesion, the chromatin remodeler ALC1 may release the chromatin barrier for efficient repair of CPD lesions through the active sliding of nucleosomes and/or the active remodeling of chromatin-bound DNA repair components, as recently shown for XRCC1 and PARP2 in single-strand break repair[47]. This may allow critical steps in GGR, such as the handover of XPC and TFIIH, a process that seems to be tightly controlled by the retention of DDB2 on chromatin[23]. We propose that the early recognition of 6-4PPs by XPC and the transient response of the XPC-PARP-ALC1 axis serves as a molecular bookmarking system, which primes chromatin containing difficult-to-repair CPDs for efficient repair at later timepoints (Fig. 9e). Together, XPC, PARP1/2, and ALC1 appear to do the heavy lifting necessary for efficient CPD repair.

## Method

**Cell lines.** All cell lines (listed in Supplementary Table 1) were cultured at 37 °C in an atmosphere of 5% CO$_2$ in DMEM (Thermo Fisher Scientific/Sigma) supplemented with penicillin/streptomycin (Sigma/GIBCO) and 10% fetal bovine serum (FBS; Bodinco BV/GIBCO).

**Plasmids.** All plasmids used are listed in Supplementary Table 2. The DDB2 gene from the mCherry-NLS-DDB2 plasmid[20] was inserted into the pcDNA5-FRT-TO-Puro-GFP-C1 plasmid[4] as an HpaI/KpnI fragment. The macroH2A1.1-GFP cassette was amplified by PCR (Supplementary Table 3) and inserted as an EcoRI/BamHI fragment into the mCherry-LacR-NLS-C1[4]. The GFP-ALC1 Δmacro-domain cassette contains an insertion with a stop codon after amino acid 726 and was inserted into pcDNA5/FRT/TO-Hygro plasmid (Invitrogen) as described for pcDNA5/FRT/TO-GFP-ALC1 WT and E175Q[47]. The PARP1-GFP (GenBank: BC037545) and GFP-PARP2 cassettes (GenBank: NM_001042618.2)[47] were amplified by PCR (Supplementary Table 3) and inserted into the pcDNA5/FRT/TO-Hygro (+NheI) plasmid as NotI/XhoI or NheI/NotI fragments, respectively. pcDNA5/FRT/TO-Hygro (+NheI) plasmid was generated by adding a NheI restriction site to the multiple cloning site with primers described in Supplementary Table 3. The XPC cDNA from XPC-EGFP[54] was digested with NrUI × NotI × NarI and ligated into pcDNA/FRT/TO/Puro digested with NotI × EcoRV.

**Generation of knockout cell lines.** To generate knockouts, U2OS (FRT) cells were co-transfected with pLV-U6g-PPB encoding a guide RNA from the LUMC/Sigma-Aldrich sgRNA library (see Supplementary Table 3 for plasmids, Supplementary Table 4 for sgRNA sequences) targeting a specific gene together with an expression vector encoding Cas9-2A-GFP (pX458; Addgene #48138) using lipofectamine 2000 (Invitrogen). Transfected U2OS (FRT) cells were selected on puromycin (1 µg/ml) for three days, plated at low density after which individual clones were isolated. Knockout clones were verified by western blot analysis.

**Generation of doxycycline-inducible cell lines.** To generate doxycycline-inducible cell lines, U2OS (FRT) cells were co-transfected with a pcDNA5/FRT/TO vector encoding the respective gene-of-interest (see Supplementary Table 2), and the pOG44 plasmid, encoding the Flp recombinase, in a 4:1 ratio according to Invitrogen's protocol of the Flp-In Core system. Cells were selected for two weeks with 50 mg/mL hygromycin B (Thermo Fisher Scientific) and expanded. The expression in U2OS (FRT) cell lines was induced with 1 mg/mL doxycycline for 24 h.

**Immunoprecipitation for Co-IP**. Cells were mock-treated or UV-C irradiated (20 J/m$^2$) and harvested after 1 h. Cell pellets were lysed for 20 min on ice in EBC-150 buffer (50 mM Tris pH 7.5, 150 mM NaCl, 0.5% NP-40, 2 mM MgCl2, protease inhibitor cocktail; Roche) supplemented with 500 U/mL Benzonase® Nuclease (Novagen). Cell lysates were incubated for 1.5 h at 4 °C with GFP-Trap®_A beads (Chromotek). The beads were then washed six times with EBC-150 buffer (50 mM Tris pH 7.5, 150 mM NaCl, 0.5% NP-40, 1 mM EDTA, protease inhibitor cocktail (Roche)) and boiled in Laemmli-SDS sample buffer.

**Immunoprecipitation to detect PARylation**. Cells were incubated with PARG inhibitor (1 µM PDD 00017273, Sigma) for 10 min and subsequently mock-treated or UV-C irradiated (50 J/m$^2$) and harvested after 10 min. Cell pellets were lysed for 60 min on ice in EBC-1 buffer (50 mM Tris pH 7.5, 150 mM NaCl, 0.5% NP-40, 2 mM MgCl2, 1 µM PARP inhibitor Olaparib; 1 µM PARG inhibitor; 1 µM, protease inhibitor cocktail; Roche) supplemented with 500 U/mL Benzonase® Nuclease (Novagen). Cleared lysates were subjected to immunoprecipitation with GFP-Trap®_A beads (Chromotek) for 1.5 h at 4 °C. The beads were then washed with EBC-2 buffer (50 mM Tris pH 7.5, 150 mM NaCl, 0.5% NP-40, 1 mM EDTA, protease inhibitor cocktail; Roche) for six times, EBC-2 buffer supplemented with 300 mM M NaCl (four times) and EBC-2 buffer supplemented with 1 M NaCl (two times). All buffers contained 1 µM PARP inhibitor (Olaparib) and 1 µM PARG inhibitor (PDD 00017273, Sigma). Finally, beads were boiled in Laemmli-SDS sample buffer and immunoprecipitated proteins were resolved by SDS-PAGE and processed for immunoblotting.

**Western blot**. Cells were spun down, washed with PBS, and boiled for 10 min in Laemmli buffer (40 mM Tris pH 6.8, 3.35% SDS, 16.5% glycerol, 0.0005% Bromophenol Blue and 0.05 M DTT). Proteins were separated on 4–12% Criterion XT Bis-Tris gels (Bio-Rad, #3450124) in NuPAGE MOPS running buffer (NP0001-02 Thermo Fisher Scientific), and blotted onto PVDF membranes (IPFL00010, EMD Millipore). The membrane was blocked with blocking buffer (Rockland, MB-070-003) for 1 h at RT. The membrane was then probed with antibodies (listed in (Supplementary Table 5) as indicated. An Odyssey CLx system (LI-COR Biosciences) was used for detection.

**Mass spectrometry sample preparation**. After pull-down, the beads were washed four times with EBC-2 buffer without NP-40 and two times with 50 mM ammonium bicarbonate followed by overnight digestion using 2.5 µg trypsin at 37 °C under constant shaking. Peptides were desalted using a Sep-Pak tC18 cartridge by washing with 0.1% acetic acid. Finally, peptides were eluted with 0.1% formic acid/ 60% acetonitrile and lyophilized as described[55].

**Mass spectrometry data acquisition**. Mass spectrometry was performed essentially as previously described[56]. Samples were analyzed on a Q-Exactive Orbitrap mass spectrometer (Thermo Scientific, Germany) coupled with an EASY-nanoLC 1000 system (Proxeon, Odense, Denmark). Digested peptides were separated using a 20 cm fused silica capillary (ID: 75 µm, OD: 375 µm, Polymicro Technologies, California, US) in-house packed with 1.9 µm C18-AQ beads (Reprosher-DE, Pur, Dr. Maisch, Ammerburch-Entringen, Germany). Peptides were separated by liquid chromatography using a gradient from 2% to 30% acetonitrile in 0.1% formic acid. For the ALC1 samples and related GFP controls the gradient length was 40 min. For the XPC, PARP1 and PARP2 samples gradient length was 100 min. Every gradient was followed by an increase to 95% acetonitrile and back to 2% acetonitrile in 0.1% formic acid for chromatography column re-conditioning. Flow rate was set to 200 nl/min for 2 h. The mass spectrometer was operated in positive-ion mode at 2.8 kV with the capillary heated to 250 °C. Data-dependent acquisition mode was used to automatically switch between full-scan MS and MS/MS scans, employing a top 7 method. Full scan MS spectra were obtained with a resolution of 70,000, a target value of $3 \times 10^6$, and a scan range of 400–2,000 m/z (XPC samples) or 300–1600 m/z (ALC1, PARP1, and PARP2 samples). Higher-Collisional Dissociation (HCD) tandem mass spectra (MS/MS) were recorded with a resolution of 35,000, a target value of $1 \times 10^5$, and normalized collision energy of 25%. The precursor ion masses selected for MS/MS analysis were subsequently dynamically excluded from MS/MS analysis for 60 s and Precursor ions with a charge state of 1 and greater than 6 were excluded from triggering MS/MS events. Maximum injection times for MS and MS/MS were 20 and 120 ms (XPC) or 250 and 120 ms (ALC1, PARP1, and PARP2), respectively.

**Mass spectrometry data analysis**. All raw data were analyzed using MaxQuant (version 1.6.6.0) as described previously[57]. We performed the search against an in silico digested UniProt reference proteome for Homo sapiens including canonical and isoform sequences (27th May 2019). Database searches were performed according to standard settings with the following modifications. Digestion with Trypsin/P was used, allowing 4 missed cleavages. Oxidation (M), Acetyl (Protein N-term) were allowed as variable modifications with a maximum number of 3. Carbamidomethyl (C) was disabled as a fixed modification. Label-Free Quantification was enabled, not allowing Fast LFQ. iBAQ was calculated. Output from MaxQuant Data was further processed on the Perseus computational platform (v 1.6.7.0)[58]. LFQ intensity values were log$_2$ transformed and potential contaminants

and proteins identified by site only or reverse peptide were removed. Samples were grouped in experimental categories and proteins not identified in 4 out of 4 replicates in at least one group were also removed. Missing values were imputed using normally distributed values with a 1.8 downshift (log$_2$) and a randomized 0.3 width (log$_2$) considering whole matrix values. Statistical analysis ($t$ tests) was performed to determine which proteins were significantly enriched in each sample compared with the others. Statistical analysis output tables were further processed in Microsoft Excel for comprehensive browsing of the datasets. Interactive Volcano plots were generated using VolcanoseR[59] and Excel.

**UV-C laser micro-irradiation**. Cells were grown on 18-mm quartz and placed in a Chamlide CMB magnetic chamber in which growth medium was replaced by CO$_2$-independent Leibovitz's L15 medium (Thermo Fisher). UV-C laser tracks were made using a diode-pumped solid-state 266 nm Yttrium Aluminum Garnet laser (Average power 5 mW, repetition rate up to 10 kHz, pulse length 1 ns). The laser is integrated into a UGA-42-Caliburn/2 L Spot Illumination system (Rapp OptoElectronic). Micro-irradiation was combined with live-cell imaging in an environmental chamber set to 37 °C on an all-quartz widefield fluorescence Zeiss Axio Observer 7 microscope, using a ×100 (1.2 NA) ultrafluar glycerol-immersion objective (UV-C). The laser system is coupled to the microscope via a triggerbox and a neutral density (ND-1) filter blocks 90% of the laser light. An HXP 120 V metal-halide lamp was used for excitation. Images were acquired in Zeiss ZEN and quantified in ImageJ.

**Chromatin expansion at site of UV-C laser damage**. Cells were seeded on 18-mm quartz coverslips and transiently transfected with 100 ng of DDB2-mCherry and 1 µg of PAGFP-H2A plasmids. The following day, cells were placed in a Chamlide CMB magnetic chamber in which the growth medium was replaced by CO$_2$-independent Leibovitz's L15 medium (Thermo Fisher). Cells were sequentially irradiated with a diode-pumped solid-state 266 nm Yttrium Aluminum Garnet laser (Average power 5 mW, repetition rate up to 10 kHz, pulse length 1 ns) to generate UV-C-specific DNA damage, immediately followed by irradiation of the same region with a diode-pumped solid-state 355 nm Yttrium Aluminum Garnet laser (average power 14 mW, repetition rate up to 200 Hz) to photo-activate PAGFP-H2A. Both lasers were integrated into a UGA-42-Caliburn/2 L Spot Illumination system (Rapp OptoElectronic). An HXP 120 V metal-halide lamp was used for excitation. Images were acquired in Zeiss ZEN and quantified in ImageJ.

**Unscheduled DNA synthesis**. 180,000 cells were seeded on 18-mm glass coverslips in 12-wells plates in DMEM with 1% FBS. After 24 h, cells were locally irradiated through a 5 µm filter with 30 J/m$^2$ UV-C. Cells were subsequently pulse-labeled with 20 µM 5-ethynyl deoxy-uridine (EdU; VWR) and 1 µM FuDR (Sigma-Aldrich) for either 1 h or 4 h. After labeling, cells were medium-chased with 10 µM thymidine in DMEM without supplements for 30 min, and fixed for 15 min with 3.7% formaldehyde in PBS. Cells were permeabilized for 20 min in PBS with 0.5% Triton-X-100 and blocked in 3% bovine serum albumin (BSA, Thermo Fisher) in PBS. The incorporated EdU was coupled to Attoazide Alexa Fluor 647 using Click-iT chemistry according to the manufacturer's instructions (Invitrogen). After coupling, the cells were post-fixed with 2% formaldehyde for 10 min and subsequently blocked with 100 mM Glycine. DNA was denatured with 0.5% NaOH for 5 min, followed by blocking with 10% BSA (Thermo Fisher) for 15 min. Next, the cells were incubated with an antibody against CPDs (see Supplementary Table 5) for 2 h, followed by secondary antibodies 1 h, and DAPI for 5 min. Cells were mounted in Polymount (Brunschwig).

**RNA recovery assay**. In all, 30,000 cells were seeded on 12 mm glass coverslips in 24-wells plates in DMEM with 1% FBS. After 24 hours, cells were irradiated with UV-C at a dose of 6 J/m$^2$ and incubated in a conditioned medium for different time periods (0, 3, and 20 hours) to allow DNA repair and to restart RNA synthesis. Following incubation, nascent RNA was labeled by incubating the cells with 400 µM 5-ethynyluridine (5-EU; Jena Bioscience; CLK-N002-10,), which was then visualized with a click-iT mix consisting of 50 mM Tris buffer pH8, 60 µM Atto Azide (ATTO-TEC; 647N-101), 4 mM CuSO4•5H2O, 10mM L-ascorbic acid (Sigma-Aldrich; A0278) and 1:1000 DAPI (ThermoFisher; D1306) for one hour. Cell were washed three times for 5 minutes with PBS, and mounted on microscope slides (Thermo Scientific) using Aqua Polymount (Polysciences, Inc. #18606).

**Clonogenic survival assays**. Cells were trypsinized, seeded at low density, and mock-treated or exposed to an increasing dose of UV light (2, 4, 6, 8 J/m$^2$ of UV-C 266 nm) or an increased dose of Illudin S (Santa cruz; sc-391575) for 72 h (30, 60, 100, 200 pg/mL). On day 10, the cells were washed with 0.9% NaCl and stained with methylene blue. Colonies of more than 20 cells were scored.

**C. elegans UV survival assays**. *C. elegans* wild-type (Bristol N2), single mutants parp-1 (ok988), parp-2 (ok344), xpc-1 (tm3886) and double mutants parp-1 (ok988); xpc-1 (tm3886) and parp-2 (ok344); xpc-1 (tm3886) were cultured and assayed as described[46,60]. For the germ cell and embryo survival assay, staged young adult animals were irradiated on empty agar plates at the indicated doses using two Philips TL-12

UV-B tubes (40 W). Following 24 h recovery on OP50 E. coli culture plates, three adult animals were allowed to lay eggs for 4 h on 6 cm plates seeded with HT115 bacteria, in quintuple for each UV-B dose. The number of hatched and unhatched (dead) eggs was counted 24 h later and the survival percentage was calculated. Results are plotted as the average of three independent experiments.

**LacO-LacR system for detecting PARylated proteins**. U2OS 2-6-3 cells containing 200 copies of a LacO-containing cassette were plated on an 18-mm glass coverslip. The next day the cells were co-transfected with lipofectamine 2000 (Invitrogen) and plasmid DNA for 6 h at 37 °C. Next the medium was replaced with DMEM +/+ and incubated overnight at 37 °C. Prior to the UV-C micro-irradiation, the medium was replaced with $CO_2$-independent Leibovitz L15 medium (Thermo Fisher Scientific) and cells were incubated with 10 μM of PARG inhibitor (Sigma) for 30 min. If indicated the cells were additionally incubated with 10 μM Olaparib. UV-C laser tracks were made using a diode-pumped solid-state 266-nm Yttrium Aluminum Garnet laser (average power 5 mW, repetition rate up to 10 kHz, and pulse length 1 ns). The UV-C laser is integrated into a UGA-42-Caliburn/2 L Spot Illumination system (Rapp OptoElectronic). Micro-irradiation was combined with live-cell imaging in an environmental chamber set to 37 °C on an all-quartz widefield fluorescence Zeiss Axio Observer 7 microscope, using a ×100 (1.2 NA) ultrafluar glycerol-immersion objective (UV-C). The laser system is coupled to the microscope via a TriggerBox, and a neutral density (ND-1) filter blocks 90% of the laser light. An HXP 120-V metal-halide lamp was used for excitation. Images were acquired in Zeiss ZEN and quantified in ImageJ.

**Immunoblot**. Immuno-dot and immuno-slotblot assays were performed as previously described[61]. DNA was extracted using DNeasy Blood & Tissue Kit (Qiagen 69504). DNA (300 ng per well, two to three technical replicates per sample) was vacuum-transferred to a nitrocellulose membrane using the Bio-Dot or Bio-Dot-SF apparatus (Bio-Rad, 1706542/5). Membranes were baked at 80 ºC in a Bio-Rad's Gel Dryer model 583, blocked in 5% milk in PBS with 0.1% Tween (PBST), washed three times in PBST, and incubated with 6-4PP or CPD antibodies (see Supplementary Table 5) overnight at 4 °C. Membranes were again washed in PBST and incubated with HRP-conjugated anti-mouse antibodies (ECL Mouse IgG, HRP-linked whole Ab (from sheep), Cytiva, NA931). Damage signal was detected using enhanced chemiluminescence (ECL™ Prime Western Blotting System, GE Healthcare, RPN2232) and exposure in the Bio-Rad ChemiDoc™ XRS + imaging system. Genomic DNA amount loaded onto the membrane was quantified using SYBR™ Gold Nucleic Acid Gel Stain (1:5000 dilution, Invitrogen, S11494), by incubating the membrane with SYBR-Gold solution for 60 min, followed by three washes with PBST. Damage signal was normalized to SYBR-Gold signal using Image Lab version 6.0 from Bio-Rad.

**Immunofluorescence microscopy**. Immunofluorescence staining was performed as described previously[47]. In brief, cells were fixed with 2% paraformaldehyde + 0.1% Triton X-100 for 15 min at room temperature, washed 3× with PBS + 0.1% Triton X-100 and subsequently permeabilized with PBS + 0.1% Triton X-100 for 2×10 min at room temperature. After blocking the cells in PBS + (PBS + 0.5% BSA + 0.15% glycine), cells were incubated with the primary antibody (see Supplementary Table 5), diluted in PBS + , overnight at 4 °C. Unspecific antibody staining was removed by washing the cells 5× in PBS + 0.1% Triton X-100. Subsequently, cells were stained with Alexa Fluor 488/568-conjugated fluorescent secondary antibody (Thermo Fisher), diluted 1:500 in PBS+, for 1 h at room temperature. Finally, cells were washed 5× with PBS + 0.1% Triton X-100, stained with Hoechst 33342 (Fisher Scientific, 1:5000 in PBS) for 10 min, and washed 3× in PBS. The immunofluorescence intensities were measured on a Zeiss Axio Observer Z1 confocal spinning-disk microscope equipped with an sCMOS ORCA Flash 4.0 camera (Hamamatsu), using a Plan-Apochromat ×40/0.95-KOrr air objective or a ×40 C-Apochromat/1.2-KOrr water objective.

**Nuclear PAR levels**. Cells were grown for 24 h in 96 well SCREENSTAR plates (Greiner Bio-One) and incubated for 10 min with 1 μM of PARG inhibitor (PDD 00017273, Sigma) before irradiation with 20 J/m² UV-C light (Stratalinker 1800, Agilent Genomics). After UV-C treatment, the cells were incubated for 5 min at 37 °C in the presence of PARG inhibitor, and subsequently fixed and stained for immunofluorescence as described above. Nuclear PAR levels were quantified in ImageJ by thresholding cell nuclei using the Hoechst signal and subsequently measuring the mean fluorescent signal of poly-(ADP-ribose) in the nucleus.

**PAR levels at local UV damage**. Cells were seeded on 18-mm glass coverslips in 12-wells plates in DMEM with 1% FBS. After 24 h, the medium was replaced with $CO_2$-independent Leibovitz L15 medium (Thermo Fisher Scientific) and cells were incubated with 10 μM of PARG inhibitor (PDD 00017273, Sigma) for 10 min. Subsequently, cells were locally irradiated with 30 J/m² UV-C (TUV PL-S 9 W; Philips) through a polycarbonate mask with pores of 5 μm (Millipore) as described[62]. After UV-C treatment, the cells were incubated for 5, 10, or 20 min at 37 °C in the presence of PARG inhibitor, and subsequently fixed and stained for immunofluorescence as described above. The mean fluorescent signal of PAR levels at sites of local damage was quantified in ImageJ by thresholding cell nuclei using

the DAPI signal and thresholding sites of local damage by using the signal of either DDB2 or XPC.

**XPC immunofluorescence at local UV-C lesions**. Cells were grown on coverslips (Fisher Scientific). Before irradiation, the coverslips were covered by 5 μm nano-pore filters (Millipore) to allow local UV-C irradiation. The cells were then irradiated with 100 J/m² UV-C light (Stratalinker, Agilent Genomics) and fixed 10 min after irradiation. The immunofluorescence was essentially done as described above, with the addition of a denaturation step with 0.07 M NaOH in PBS for 5 min and a second blocking step in PBS + before incubation with the primary antibody, to allow recognition of CPD lesions. The enrichment of XPC at CPD lesions was quantified in ImageJ. Nuclei and CPD lesions were recognized by thresholding the Hoechst and CPD signal, respectively. The mean fluorescent intensity of XPC and CPD was measured at CPD spots and in the rest of the nucleus. The enrichment of XPC/CPD at UV-C lesions was quantified as followed: mean fluorescence(spot)/mean fluorescence(nucleus background)−1.

**Proteins**. Recombinant XPC-RAD23B was expressed in Sf9 insect cells and purified as described previously[63]. XPC has 3xFLAG tag at the N-terminus.

**In vitro PARylation**. In vitro PARylation was performed in 10 μl volume at 25 °C for 30 min as described previously[64]. The PARylation mixture contained buffer (100 mM Tris-HCl pH8.0, 10 mM MgCl2, 10% glycerol, 1.5 mM DTT, 100 μg/ml BSA, 20 μM NAD and 1 pmole of PARP1 protein. XPC-RAD23B was added at 0, 0.25, 0.5, and 1 pmole for every pmole of PARP1. The tubes containing the above ingredients were incubated at 25 °C for 5 min. Following this, 0.5 pmoles of plasmid DNA irradiated with 5000 J/m² of UV-C was added to each tube and they were incubated further for 30 min. The reaction was stopped by adding an equal volume of 2× Laemmli buffer and the samples were separated on denaturing 6% SDS-PAGE. The gel was transferred on nitro-cellulose and probed for PAR after which blots were additionally proved for PARP1 and XPC. The blots were revealed using chemiluminescence and the ChemiGenius2 Bioimaging system. Images captured under non-saturating conditions were used to analyze and estimate the PAR signal using the GeneSnap 6.0 sofware.

**Reporting summary**. Further information on research design is available in the Nature Research Reporting Summary linked to this article.

## Data availability
The raw mass spectrometry proteomics data generated in this study have been deposited in the ProteomeXchange Consortium via the PRIDE[65] partner repository under accession code PXD025226. Source data are provided with this paper.

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

## Acknowledgements

The authors acknowledge Sylvie Noordermeer for the PARP inhibitor olaparib. We thank Artur Mayerhofer for kindly providing the Stratalinker UV device for irradiation. We thank Nicholas Lakin for providing PARP1-KO and PARP2-KO cells. We thank Susan Janicki for providing U2OS 2-6-3 cells. *C. elegans* strains were provided by the Caenorhabditis Genetics Center (funded by NIH Office of Research Infrastructure

Programs P40 OD010440). We thank Masanao Miwa (National Cancer Center Research Institute, Tokyo) for providing 10H hybridoma cells obtained through the Riken cell bank. We thank Mihaela Robu for feedback on the manuscript. This research was supported by an LUMC Research Fellowship, ENW-M (OCENW.KLEIN.090), and ALW-VIDI grants (ALW.016.161.320) from the Dutch Research Council (NWO) to M.S.L. This research was further funded by the DFG (German Research Foundation) through Project-ID 213249687 - SFB 1064 and Project-ID 325871075 - SFB 1309, as well as LMU to A.G.L. C.B. was the recipient of a grant from the Stiftungskommission of the LMU Medical Faculty. R.G-P was the recipient of a Young Investigator Grant from the Dutch Cancer Society (KWF-YIG 11367). A.C.O.V. was funded by an ERC grant (310913). H.L. and M.v.d.W. were funded by the Netherlands Organization for Scientific Research (711.018.007 and CancerGenomiCs.nl) and the Oncode Institute, which is partly financed by the Dutch Cancer Society. O.D.S. was supported by the Korean Institute of Basic Science (IBS-R022-A1) and the National Cancer Insitute (USA, P01-CA092584). G.M.S. was supported by grants from the Research Centre of CHU de Quebec Laval University as well as from the Natural Sciences and Engineering Research Council of Canada through the Discovery Grant (RGPIN-2016-05868) and the Discovery Accelerator Supplement Grant (RGPAS-492875-2016). S.A. was funded by the Israel Cancer Research Fund Research Career Development Award (3013004741), the Israel Cancer Association grant (20210078), and Israel Science Foundation grant (1710/17) administered by the Israeli Academy of Science and Humanities and is the recipient of the Jacob and Lena Joels memorial senior lectureship. H.v.A. was funded by a VICI grant from the Dutch Research Council (NWO-VICI grant VI.C.182.052).

## Author contributions

C.B. generated plasmids, U2OS Flp-In cell lines, performed clonogenic survivals, co-IP experiments, western blot analyses, UV-C laser micro-irradiation experiments, local damage, and nuclear PAR immunofluorescence stainings. KA generated and validated U2OS single-KO cells, generated plasmids and U2OS Flp-In cell lines, performed co-IP experiments for western blot and mass spectrometry, UDS experiments, western blot analyses, UV-C laser micro-irradiation, LacR-based PARylation assays and prepared genomic DNA for immunoblot experiments. D.v.d.H. performed all PAR immunofluorescence at sites of local damage, C.G-L., T.T.K., and J.C. performed denaturing IP experiments, M.B.R. and H.v.A. performed chromatin expansion assays, M.v.d.W. and H.L. performed GGR sensitivity assays in *C. elegans*. R.G-P analyzed the mass spectrometry samples with support from A.C.O.V. A.Y., A.P., S.A. performed and analyzed all immunoblot experiments. D.E.C.B. performed clonogenic survivals and western blot analyses. A.K. performed RRS experiments and performed clonogenic Illudin S survivals. R.G.S. and G.M.S. performed the in vitro PARylation assay. H.S.K. and O.D.S. provided recombinant XPC-RAD23B complex. A.G.L. and M.S.L. conceived, coordinated, and supervised the project. C.B. and M.S.L. drafted the manuscript. All authors commented and edited the manuscript.

## Competing interests

A.G.L. is a founder, shareholder, and managing director of Eisbach Bio, a biotech developing small molecule therapeutics. The remaining authors declare no competing interests.
