## [Peer Review File · Nature Communications]

XPC–PARP complexes engage the chromatin remodeler ALC1 to catalyze global genome DNA damage repairREVIEWER COMMENTS

Reviewer #1 (Remarks to the Author):

This manuscript describes intriguing findings concerning physical and functional interactions between XPC, the DNA damage recognition factor initiating global genome nucleotide excision repair (GG-NER), and poly (ADP-ribose) polymerases PARP1/2. First, the authors provide with evidence for the XPC-PARP interactions from proteomic and co-IP experiments. It is notable that not only XPC, but also the poly (ADP-ribose) (PAR)-related chromatin remodeler ALC1, seem to interact with PARP2 much more strongly than PARP1 in a DNA damage-independent manner (Fig. 2e, f). Despite such constitutive interactions, recruitment of XPC and PARPs to DNA damage does not depend on each other (Fig. 3). Next the authors show that UV irradiation rapidly induces PARylation within the entire nucleus, which is enhanced by XPC (Fig. 4e). PARP1 activity in vitro is indeed stimulated by the presence of XPC (Fig. 4g, h), while the LacO tethering experiments suggest the possibility that XPC itself could be PARylated in a DNA damage-dependent manner (Fig. 4c). Loss of PARP1 or PARP2 reduces repair rates of UV-induced CPDs (Fig. 5b, c) and sensitizes human cells as well as *C. elegans* to UV (Fig. 5a, d). In addition, they identify YBX1 as a novel PARP interactor, which seems to be required for UV-induced PARylation activity (Fig. 5f). Finally the authors focus on ALC1, which is rapidly recruited to DNA damage depending on its macrodomain and the presence of XPC (Fig. 6d). On the other hand, loss of ALC1 does not affect recruitment of XPC or DDB2, but partially compromises UV survival of cells and removal of UV-induced DNA damage from the genome (Fig. 7, 8). Taken together, the authors conclude that XPC and PARPs recruited to DNA damage induce PARylation, which then facilitates recruitment of ALC1 and the following NER process.

Overall the experiments are well designed and conducted in quite high quality, which this reviewer highly evaluates. Although the manuscript is mostly well written, conclusions made by the authors are not sufficiently supported by the presented results. This reviewer would raise several concerns, which should be addressed before publication.

1. The authors first show the evidence that XPC and ALC1 interact with PARP2 much more robustly than PARP1. However, in later part, roles of the two PARPs seem quite redundant, which makes relevance of the PARP2 specificity very difficult to interpret. At least they should discuss possible functional difference of the PARPs.
2. Fig. 3: Recruitment of XPC and PARPs to DNA damage seems mutually independent. Given that the proteomic and co-IP experiments are performed with soluble protein fractions, but not DNA-bound forms, it is also a bit difficult to understand relevance of the XPC-PARP interactions. Could the authors assess which percentage of XPC and/or PARPs in a cell is present as a complex?
3. Fig. 4a: The authors show that global UV irradiation induces PARylation throughout the nucleus. If local UV irradiation was adopted, is PARylation induced specifically at DNA damage? If so, does it depend on XPC or DDB2? They show recruitment of PARP1/2 themselves (Fig. 3), while this does not necessarily mean induction of PARylation.
4. Fig. 4a: It is also a bit curious that PARP2-KO cells still exhibit substantial levels of UV-induced PARylation, while these cells are more sensitive to UV than PARP1-KO cells. This raises the possibility that only a minor fraction of the UV-induced PARylation is relevant to GG-NER. Although loss of PARP1 severely reduces overall PARylation, residual activity could be limited more specifically to DNA damage sites.
5. Fig. 4d: Although this is a quite sophisticated and interesting experiment, interpretation of the results should be made more carefully. For instance, the presented data do not exclude the possibility that XPC is recruited to the LacO array through interactions with other PARylated proteins.

6. Fig. 4g: This data also does not support the conclusion that XPC is PARylated, because the observed PAR signals do not tell which is PARylated, XPC or PARP1. It is necessary to show band shifts of XPC caused by PARylation.

7. Fig. 5: The authors could consider to address epistatic relationships between XPC and PARPs with human cells and *C. elegans*. It would be relevant to know to which extent the functions of PARP1/2 are related to GG-NER.

8. Fig. 6f: These data do not exclude the possibility that ALC1 is recruited to DNA damage depending on later NER steps. Because PARPs are known to be activated by binding to DNA ends, involvement of dual incisions in UV-induced PARylation may need to be considered throughout this manuscript.

9. Fig. 8: Theoretically XPC-KO cells must be almost fully inactive in removal of 6-4PPs or CPDs. These data, therefore, suggest the possibility of 'lesion dilution' by DNA replication, while it is not described in materials & methods how this adverse effect is corrected. Particularly if UV irradiation differentially affects growth of these cell lines, it would compromise reliability of the data substantially.

Reviewer #2 (Remarks to the Author):

This paper provides new insight into how chromatin organization impacts repair of UV lesions by GGR and related pathways. The authors identify a direct interaction between XPC and PARPs 1 and 2 which is independent of DNA damage. Further, they demonstrate PARylation of XPC. Importantly, this leads to recruitment of Alc1, a remodeler which is then required for repair of CPDs. The experimental approach is robust, utilizing proteomics, live cell imaging and genetic approaches. The use of the LacR-macro domain to identify PARylated proteins is an excellent approach. A key observation is that PARP2 interacts more strongly with ALC1-XPC than PARP1. However, PARPs and XPC appear to be recruited separately and independently to UV lesions. Further studies showing that Alc1 recruitment depends on XPC and PARP provides an important link to potential chromatin remodeling during UV repair. This might be relevant to the "difficult to repair lesions" discussed in the paper. Overall, the paper provides strong evidence for new roles for PARP1 and/or PARP2 and Alc1 in repairing a subset of UV lesions, and helps to unravel the complex and distinct contributions of PARP1 vs PARP2 to DNA repair. Some additional comments are outlined below.

1. It would be useful to show PARylation of e.g. Alc1 and PARPs 1 and 2 after UV by western blot to confirm this modification (figure 4) and to test the proposal (line 195) that PARP2 may either exhibit rapid PAR turnover or is not extensively PARylated. This should be extended to XPC and XPC KO cell lines.

2. The in vitro data showing that PARP1 activity is increased by XPC is strong, but it would be important to strengthen this with cell based assays.

3. In Fig 5B, 5C, the differences are small and the SEMs overlap at certain time points. It would be helpful to provide statistical analysis of the significance of the differences shown at each time point. This is particularly important to support the conclusion (line 222-223) that PARP2 had a more significant defect than PARP1, since these repair time courses look identical. In this case, it would be unclear why cells lacking PARP2 are more UV sensitive (Fig 5a) but have the same repair defect as PARP1 KO. In the *C. elegans* data, PARP1 and 2 loss have the same impact on UV survival.

4. The data on YBX1 is interesting, but, on its own, does not really add to the paper. A systematic analysis of all the proteins identified by MS for their impact might be more informative here.

5. Although proteomics and IP studies show interactions between PARPs and e.g. XPC, because these proteins are recruited independently to lesions, the role of the observed interaction remains unclear. Histones were frequently identified in the MS analysis. Do the conditions used for IP and MS preserve nucleosomal integrity (e.g. use of benzonase)? The observed interactions may reflect binding to the nucleosome, rather than direct protein-protein interactions.

6. In figure 6c, levels of ALC1E175Q expression are lower than wt. This may explain the apparent reduction in PARP1 and PARP2 interaction. Quantifying the ratio of alc1/PARP1 and 2 would help address this. The apparent reduced interaction of PARP with the Alc1/macro deletion might be because this mutant fails to localize to chromatin/nucleosomes (fig 6d).
7. The data showing requirement for Alc1 and its ATPase domain in UV repair is excellent, providing strong evidence for linkage between XPC and Alc1 function. Does the decreased survival of the Alc1E175Q lines relative to KO suggest that the increased residence time of this complex at lesions (Fig 6E) further blocks repair? Does this lead to altered/extended PAR levels due to reduced repair or altered repair kinetics (fig 8b and 8d)?
8. The authors frequently refer to Alc1 as promoting "active chromatin remodeling (line 378). Can the authors provide some experimental evidence that Alc1 is actually altering chromatin/nucleosome function at UV lesions? What activity of Alc1 (sliding? ejection of nucs) would be needed? An dis this regulated by PARYlation or PARP2?

Thank you and the referees for the thoughtful and useful comments, which we have taken fully on-board and have addressed experimentally to the full extent, both through a range of new experiments addressing the important and valuable suggestions, as well as through a revised manuscript and improved discussion.

Below you find a point-by-point response to the referee's comments.

For your orientation, the referee's original points are in black, while our responses are in red.

We thank the referees for their refereeing of our revised manuscript and thank you in advance for your additional guidance. We look forward to hearing back from you in due course.

With best regards, yours,

Martijn Luijsterburg and Andreas Ladurner, on behalf of all co-authors

Individual Responses to Reviewer comments:

Reviewer #1:

This manuscript describes intriguing findings concerning physical and functional interactions between XPC, the DNA damage recognition factor initiating global genome nucleotide excision repair (GG-NER), and poly (ADP-ribose) polymerases PARP1/2. First, the authors provide with evidence for the XPC-PARP interactions from proteomic and co-IP experiments. It is notable that not only XPC, but also the poly (ADP-ribose) (PAR)-related chromatin remodeler ALC1, seem to interact with PARP2 much more strongly than PARP1 in a DNA damage-independent manner (Fig. 2e, f). Despite such constitutive interactions, recruitment of XPC and PARPs to DNA damage does not depend on each other (Fig. 3). Next the authors show that UV irradiation rapidly induces PARylation within the entire nucleus, which is enhanced by XPC (Fig. 4e). PARP1 activity in vitro is indeed stimulated by the presence of XPC (Fig. 4g, h), while the LacO tethering experiments suggest the possibility that XPC itself could be PARylated in a DNA damage-dependent manner (Fig. 4c). Loss of PARP1 or PARP2 reduces repair rates of UV-induced CPDs (Fig. 5b, c) and sensitizes human cells as well as *C. elegans* to UV (Fig. 5a, d). In addition, they identify YBX1 as a novel PARP interactor, which seems to be required for UV-induced PARylation activity (Fig. 5f). Finally, the authors focus on ALC1, which is rapidly recruited to DNA damage depending on its macrodomain and the presence of XPC (Fig. 6d). On the other hand, loss of ALC1 does not affect recruitment of XPC or DDB2, but partially compromises UV survival of cells and removal of UV-induced DNA damage from the genome (Fig. 7, 8). Taken together, the authors conclude that XPC and PARPs recruited to DNA damage induce PARylation, which then facilitates recruitment of ALC1 and the following NER process.

Overall, the experiments are well designed and conducted in quite high quality, which this reviewer highly evaluates. Although the manuscript is mostly well written, conclusions made by the authors are not sufficiently supported by the presented results. This reviewer would raise several concerns, which should be addressed before publication.

We thank the reviewer for their interest, excitement and constructive feedback. Thank you for suggesting experiments and clarifications that have improved our experimental manuscript. We have addressed all points below. New experiments were conducted that address the specific questions posed. Thank you also in advance for your time in assessing our revised manuscript.

1. The authors first show the evidence that XPC and ALC1 interact with PARP2 much more robustly than PARP1. However, in later part, roles of the two PARPs seem quite redundant, which makes relevance of the PARP2 specificity very difficult to interpret. At least they should discuss possible functional difference of the PARPs.

Indeed, at the biochemical level the interactions with PARP2 appear to be stronger. We thank the referee for encouraging us to discuss potential differences in PARP1 vs. PARP2. In the Discussion, we now speculate on the functional difference of the PARPs (**line 376 - 392**), as follows:

“In several DNA repair pathways, PARP1 is accompanied by a second poly-(ADP-ribose) polymerase, PARP2. The coordinated action of both PARP1 and PARP2 seems to be required for efficient base excision repair and DNA double-strand break repair (Caron et al., 2019; Fouquin et al., 2017; Ronson et al., 2018). However, the precise function of PARP2 has so far largely remained elusive and has not yet been described in nucleotide excision repair. Here, we identify PARP2 as a novel regulator of the GGR response. The protein displays abundant interactions with XPC and especially with the chromatin remodeler ALC1, suggesting that it is tightly linked to the newly identified XPC-PARP axis. Interestingly, while the contribution of PARP2 to the UV-induced PAR response was minor, we found that PARP2 deletion strongly sensitized cells to UV and was important for the repair of difficult-to-repair CPD lesions. This sparks the question whether PARP2 may contribute to GGR independent of its catalytic activity. Such a mechanism of regulation was proposed previously for the efficient repair of DNA double-strand breaks by homologous recombination, where PARP2 limits 53BP1 accumulation and promotes end-resection independently of its catalytic activity (Fouquin et al., 2017). Alternatively, PARP2 may contribute to the synthesis of distinct PAR chains, such as branched PAR molecules at UV-C lesions (Chen et al., 2018). Smaller quantities of branched PAR chains may be necessary to promote efficient GGR by virtue of their recognition by specific DNA repair factors, including potentially ALC1.”

2. Fig. 3: Recruitment of XPC and PARPs to DNA damage seems mutually independent. Given that the proteomic and co-IP experiments are performed with soluble protein fractions, but not DNA-bound forms, it is also a bit difficult to understand relevance of the XPC-PARP interactions. Could the authors assess which percentage of XPC and/or PARPs in a cell is present as a complex?

We agree. In **Fig 2e**, we report iBAQ values to quantify the stoichiometry between PARP1 (100%) and XPC (0.006%), as well as PARP2 (100%) and XPC (0.07%). Overall, our findings thus indicate that the stoichiometry is quite low, meaning that only a relatively small fraction of the cellular PARP1 and PARP2 enzyme associate with XPC under the tested experimental conditions. We explain in **line 122 – 126** that:

“Intensity-based absolute quantification (iBAQ) of protein amounts indicated that ~15% of the isolated PARP2 molecules associated with ALC1, while only 0.07% of PARP1 molecules interacted with the remodeler. Additionally, the fraction of PARP2 molecules that associated with XPC was ten-fold higher than for PARP1 (Figure 2e).”

3. Fig. 4a: The authors show that global UV irradiation induces PARylation throughout the nucleus. If local UV irradiation was adopted, is PARylation induced specifically at DNA damage? If so, does it depend on XPC or DDB2? They show recruitment of PARP1/2 themselves (Fig. 3), while this does not necessarily mean induction of PARylation.

We thank the referee for this suggestion. We have now locally irradiated cells with UV-C light through 5 μ m pores, which resulted in the local accumulation of PAR signal at the sites DNA damage sites that were marked by the local accumulation of repair proteins (either DDB2 or XPC). We found that the local UV-induced PAR signal is fully dependent on PARP1 (**Fig 4a, b**), attenuated in XPC-deficient cells (~50%; **Fig 4e, f**) and not affected by loss of DDB2 (**Fig 4g, h**). These results confirm our earlier results obtained after global UV irradiation (shown in **Fig 4c, d, i, j** of the revised manuscript; see **line 161 - 176**). Our results with both local and global UV irradiation support our conclusions that XPC stimulates the PAR response at UV lesions, which is largely driven by PARP1.

4. Fig. 4a: It is also a bit curious that PARP2-KO cells still exhibit substantial levels of UV-induced PARylation, while these cells are more sensitive to UV than PARP1-KO cells. This raises the possibility that only a minor fraction of the UV-induced PARylation is relevant to GG-NER. Although loss of PARP1 severely reduces overall PARylation, residual activity could be limited more specifically to DNA damage sites.

Thank you for this comment. We show in **Fig 4a, b** that PARP2-KO cells mount a normal PAR response at sites of local UV damage, similar to wild-type cells. Further, the PAR response at sites of local DNA damage is fully dependent on PARP1 (**Fig 4a, b; line 160 - 169**).

These findings suggest that PARP1 impacts GGR by mediating the local PAR response, while PARP2 may contribute an additional layer of regulation either independently of its catalytic activity or by introducing distinct PAR species, such as branched PAR chains, that cannot be quantified well by immunofluorescence quantification of the PARylation signal. We therefore speculate on the function of PARP2 in the discussion of the manuscript (line **383 - 392**):

“Interestingly, while the contribution of PARP2 to the UV-induced PAR response was minor, we found that PARP2 deletion strongly sensitized cells to UV and was important for the repair of difficult-to-repair CPD lesions. This sparks the question whether PARP2 may contribute to GGR independent of its catalytic activity. Such a mechanism of regulation was proposed previously for the efficient repair of double-strand breaks by homologous recombination, where PARP2 limits 53BP1 accumulation and promotes end-resection independently of its catalytic activity (Fouquin et al., 2017). Alternatively, PARP2 may contribute to the synthesis of distinct PAR chains, such as branched PAR molecules at UV-C lesions (Chen et al., 2018). Smaller quantities of branched PAR chains may be necessary to promote efficient GGR by virtue of their recognition by specific DNA repair factors, including potentially ALC1.”

5. Fig. 4d: Although this is a quite sophisticated and interesting experiment, interpretation of the results should be made more carefully. For instance, the presented data do not exclude the possibility that XPC is recruited to the LacO array through interactions with other PARylated proteins.

We thank the referee for their caution. We have performed pull-down experiment under high-salt conditions to detect the UV-induced PARylation of specific proteins. While PARP1 and ALC1 become strongly PARylated in response to cellular UV irradiation (**Fig 5a, d**), XPC and PARP2 already show a basal level of PARylation in unirradiated cells, which does not increase after UV irradiation (**Fig 5b, c**). This evidence is consistent with XPC being recruited to the LacO array through its interaction with PARylated PARP1 (**line 200 - 215**).

6. Fig. 4g: This data also does not support the conclusion that XPC is PARylated, because the observed PAR signals do not tell which is PARylated, XPC or PARP1. It is necessary to show band shifts of XPC caused by PARylation.

Thank you. Our new pull-down experiments indeed show that the basal PARylation of XPC and PARP2 is not enhanced in response to UV irradiation (**Fig 5b, c**), while PARP1 and ALC1 were robustly PARylated in response to UV (**Fig 5a, d**). We revised the text according to these new results (**line 189 - 196**).

7. Fig. 5: The authors could consider to address epistatic relationships between XPC and PARPs with human cells and *C. elegans*. It would be relevant to know to which extent the functions of PARP1/2 are related to GG-NER.

This is an interesting suggestion. To understand whether the functions of PARP1/2 are related to GGR, we knocked-down XPC using siRNAs in human cells and generated XPC and PARP1 or PARP2 double knockout *C. elegans* to test their respective sensitivity to UV-light. The knock-down of XPC did not cause an additive sensitivity in either PARP1-KO or PARP2-KO cells compared to wild-type U2OS cells (**Fig 53b, c**). The same was true for XPC-PARP1 or XPC-PARP2 dKO worms (**Fig 6c, d, e**). These data thus support the conclusion that the function of PARP1 and PARP2 in the UV response is related, to a large extent, to GGR and that this relationship is evolutionarily conserved from *C. elegans* to humans (**line 230 – 247**).

8. Fig. 6f: These data do not exclude the possibility that ALC1 is recruited to DNA damage depending on later NER steps. Because PARPs are known to be activated by binding to DNA ends, involvement of dual incisions in UV-induced PARylation may need to be considered throughout this manuscript.

During this revision, we investigated the effect of XPA depletion on PARylation levels. Knockdown of XPA also caused a reduction in PAR signal at sites of local UV irradiation, although not to the same extent as knockout of XPC (**Fig S2f, g, h, i**). Furthermore, the depletion of XPA in XPC-KO cells did not result in an additive reduction of PAR levels, suggesting the early PAR response at UV lesions is mainly driven by GGR (**line 176 - 178**).

We agree with the reviewer that ALC1 may also act at later stages of GGR, but the rapid recruitment within 30 seconds (**Fig 7d, e**) and the stimulatory impact of XPC on PARP1-dependent PARylation *in vitro* (**Fig 4k, l**), suggests to us that ALC1 recruitment is initially dependent on early damage recognition.

9. Fig. 8: Theoretically XPC-KO cells must be almost fully inactive in removal of 6-4PPs or CPDs. These data, therefore, suggest the possibility of 'lesion dilution' by DNA replication, while it is not described in materials & methods how this adverse effect is corrected. Particularly if UV irradiation differentially affects growth of these cell lines, it would compromise reliability of the data substantially.

We thank the reviewers for this comment. The removal of 6-4PPs in our assays was measured within 4h, which rules out a significant dilution of DNA lesions due to replication within this relatively short time-frame. It is possible that TCR contributes to ~30% of the repair of 6-4PPs that we can detect in GGR-deficient XPC-KO cells. Alternatively, other technical reasons cause this apparent reduction within 4 h. Importantly, wild-type cells repair ~80% of the 6-4PPs during the same timeframe. This gives us a clear window to measure 6-4PP repair (**Fig 8a**). Dilutions of CPDs is certainly possible, since we measure genomic CPDs levels at 48h or 72h, which is why we included XPC-KO cells as an internal control. The ALC1-KO cells are almost as defective in CPD repair at 72h as the XPC-KO cells (**Fig 8b**), suggesting that we may underestimate the true impact of ALC1 on the kinetics of CPD repair considering that fully GGR-deficient XPC-KO cells show a very similar phenotype.

We thank the reviewer very much for their time, constructive criticism and for this re-review.

Reviewer #2:

This paper provides new insight into how chromatin organization impacts repair of UV lesions by GGR and related pathways. The authors identify a direct interaction between XPC and PARPs 1 and 2 which is independent of DNA damage. Further, they demonstrate PARylation of XPC. Importantly, this leads to recruitment of Alc1, a remodeler which is then required for repair of CPDs. The experimental approach is robust, utilizing proteomics, live cell imaging and genetic approaches. The use of the LacR-macro domain to identify PARylated proteins is an excellent approach. A key observation is that PARP2 interacts more strongly with ALC1-XPC than PARP1. However, PARPs and XPC appear to be recruited separately and independently to UV lesions. Further studies showing that Alc1 recruitment depends on XPC and PARP provides an important link to potential chromatin remodeling during UV repair. This might be relevant to the “difficult to repair lesions” discussed in the paper. Overall, the paper provides strong evidence for new roles for PARP1 and/or PARP2 and Alc1 in repairing a subset of UV lesions, and helps to unravel the complex and distinct contributions of PARP1 vs PARP2 to DNA repair. Some additional comments are outlined below.

We thank the reviewer for their interest, for finding our data to be strong and for their constructive feedback. Thank you for suggesting experiments and clarifications that have improved our experimental manuscript. We have addressed all points below. New experiments were conducted that address the specific questions posed. Thank you also in advance for your time in assessing our revised manuscript.

1. It would be useful to show PARylation of e.g. Alc1 and PARPs 1 and 2 after UV by western blot to confirm this modification (figure 4) and to test the proposal (line 195) that PARP2 may either exhibit rapid PAR turnover or is not extensively PARylated. This should be extended to XPC and XPC KO cell lines.

Thank you. We have performed pull-down experiment under high-salt conditions to detect UV-induced PARylation of specific proteins. While PARP1 and ALC1 become strongly PARylated in response to UV irradiation (Fig 5a, d), XPC and PARP2 already show a basal level of PARylation in unirradiated cells, which does not increase after UV irradiation (Fig 5b, c).

Further, we have also attempted to monitor PARylation of these proteins in XPC-KO cells as well, but these experiments were inconclusive. This was possibly the result of unequal ectopic expression of the GFP-tagged proteins in these different genetic backgrounds, making a direct comparison difficult. However, we show in our revised manuscript that XPC-KO cells have reduced PAR levels at sites of local damage (Fig 4e, f), as well as global UV irradiation (Fig 4i, j). Further, we found that XPC stimulates PARP1-dependent PARylation in a recombinant system (Fig 4k, l).

2. The *in vitro* data showing that PARP1 activity is increased by XPC is strong, but it would be important to strengthen this with cell based assays.

Thank you for the suggestion. We now show in Fig 4e, f that knockout of XPC attenuated cellular PAR levels at the site of local UV damage (~50%). Further, monitoring nuclear PAR levels after global UV irradiation in wild-type and XPC-KO cells showed a similar result, revealing a dampened PAR signal (Fig 4i, j; see line 173 – 178), suggesting that XPC does not only impact the activity of PARP1 *in vitro*, but also in cells upon UV irradiation.

3. In Fig 5B, 5C, the differences are small and the SEMs overlap at certain time points. It would be helpful to provide statistical analysis of the significance of the differences shown at each time point. This is particularly important to support the conclusion (line 222-223) that PARP2 had a more significant defect than PARP1, since these repair time courses look identical. In this case, it would be unclear why cells lacking PARP2 are more UV sensitive (Fig 5a) but have the same repair defect as PARP1 KO. In the *C. elegans* data, PARP1 and 2 loss have the same impact on UV survival.

The difference between the CPD repair defect in PARP1-KO and PARP2-KO cells is not statistically significant (shown in Fig S3d, e in our revised manuscript). We therefore toned down the conclusion in the text (line 233 - 237).

4. The data on YBX1 is interesting, but, on its own, does not really add to the paper. A systematic analysis of all the proteins identified by MS for their impact might be more informative here.

We thank the referee for this suggestion. After considering this comment within the authorship team, we have decided to concur with the referee's suggestion and have thus opted to remove the YBX1 data from the current manuscript. Indeed, the findings do not add much to the current story. We will thus work out the role of YBX1 as a putative co-factor of PARP1 separately. Regarding the systematic analysis of ALL interactors of PARP1 and PARP2 for their impact in NER, this is obviously also an interesting point. However, we do feel that this would be beyond the scope of the current manuscript, and similar to YBX1, deserves an interesting project on its own (or together with the YBX1 interactor).

5. Although proteomics and IP studies show interactions between PARPs and e.g. XPC, because these proteins are recruited independently to lesions, the role of the observed interaction remains unclear. Histones were frequently identified in the MS analysis. Do the conditions used for IP and MS preserve nucleosomal integrity (e.g. use of benzonase)? The observed interactions may reflect binding to the nucleosome, rather than direct protein-protein interactions.

Thank you for the question. Chromatin structure integrity is indeed disrupted under our conditions by the use of benzonase. This treatment results in mono-nucleosomes, as analyzed and judged by DNA fragment length analysis on agarose gels.

6. In figure 6c, levels of ALC1E175Q expression are lower than wt. This may explain the apparent reduction in PARP1 and PARP2 interaction. Quantifying the ratio of alc1/PARP1 and 2 would help address this. The apparent reduced interaction of PARP with the Alc1/macro deletion might be because this mutant fails to localize to chromatin/nucleosomes (fig 6d).

Thank you. In **Fig 7b**, we used quantitative proteomics to compare the interactome of ALC1-WT compared to ALC1-E175Q. Please note that we indicated ALC1 levels in the volcano plots and find that PARP2 clearly interacts less with the ALC1-E175Q mutant compared with ALC1-WT. These data show that the decreased interaction with PARP2 does not appear to be caused by the lower expression of ALC1-E175Q, while the decreased interaction of PARP1 with the ALC1-E175Q mutant might be attributable to differences in wild-type and mutant expression levels. We fully agree that the ALC1 Δ macro construct fails to interact with PARP1, which in turn will disrupt its ability to associate with chromatin.

7. The data showing requirement for Alc1 and its ATPase domain in UV repair is excellent, providing strong evidence for linkage between XPC and Alc1 function. Does the decreased survival of the Alc1E175Q lines relative to KO suggest that the increased residence time of this complex at lesions (Fig 6E) further blocks repair? Does this lead to altered/extended PAR levels due to reduced repair or altered repair kinetics (fig 8b and 8d)?

Thank you for the enthusiasm here. To better understand the impact of the ATPase inactive ALC1, we monitored the PAR response at sites of local UV damage at different time-points after irradiation during this revision. Wild-type cells showed clear PAR signal at sites of UV-induced lesions marked by the local enrichment of XPC, which was similar between all time-points examined (**Figures 9c, d**). In contrast, although ALC1-KO cells initially mounted a similar PAR response shortly after UV irradiation, PAR levels steadily increased over time to ~2-fold higher levels at 30 min compared to wild-type cells (**Figures 9c, d**). This hyper-PAR response could be fully rescued by expression of ALC1-WT, while expression of catalytically inactive ALC1-E175Q even further increased PAR levels to ~3-fold over wild-type cells (**Figures 9c, d**). These findings show that ALC1's catalytic activity is required to shut-off the PAR response to UV lesions, which is mounted by PARP1 and stimulated by XPC (**line 336 - 347**). ALC1 activity thus impacts the PAR response and DNA repair of UV-induced DNA lesions.

8. The authors frequently refer to Alc1 as promoting “active chromatin remodeling (line 378). Can the authors provide some experimental evidence that Alc1 is actually altering chromatin/nucleosome function at UV lesions? What activity of Alc1 (sliding? ejection of nucs) would be needed? An this this regulated by PARylation or PARP2?

We thank the referee for this suggestion. To directly investigate the impact of ALC1 chromatin remodeling at UV lesions, we established a new assay to measure chromatin expansion at UV-induced micro-irradiation sites during this revision. In **Fig 9a, b**, we sequentially irradiated cells expressing photoactivatable GFP fused to histone H2A (PAGFP-H2A) with a UV-C laser (266 nm) to generate UV-specific photolesions, and with a UV-A laser (365 nm) irradiation to activate PAGFP-H2A specifically at sites of local UV damage. While wild-type and PARP2-KO cells showed considerable expansion of PAGFP-H2A tracks following sequential UV-C and UV-A laser irradiation, such an expansion was attenuated in either PARP1-KO or ALC1-KO cells. These findings reveal that chromatin expansion at sites of UV-C-induced DNA damage is stimulated by PARP1-dependent and ALC1-mediated chromatin remodeling (**Figures 9a, b**). See **line 310 - 328**.

We thank the reviewer very much for their time and constructive criticism.

REVIEWERS' COMMENTS

Reviewer #1 (Remarks to the Author):

In this revised version of the manuscript, the authors addressed this reviewer's concerns in a satisfactory manner. This reviewer highly appreciates their tremendous efforts to add new data and revise the manuscript.

Reviewer #2 (Remarks to the Author):

The authors have done an excellent job addressing the key issues raised by the reviewers. New data and discussion provide clarification for issues and have improved the strength of the conclusions. I do not see any other major issues which need to be addressed and can recommend publication.

Point-to-point response

We thank the reviewers for their support. No additional points were raised. The manuscript was modified according to the editorial policies.